# Functional Devices from Bottom-Up Silicon Nanowires: A Review

**DOI:** 10.3390/nano12071043

**Published:** 2022-03-22

**Authors:** Tabassom Arjmand, Maxime Legallais, Thi Thu Thuy Nguyen, Pauline Serre, Monica Vallejo-Perez, Fanny Morisot, Bassem Salem, Céline Ternon

**Affiliations:** 1Univ. Grenoble Alpes, CNRS, Grenoble INP (Institute of Engineering Univ. Grenoble Alpes), LMGP, F-38000 Grenoble, France; tabassom.arjmand@grenoble-inp.fr (T.A.); maxime.legallais@cea.fr (M.L.); thuthuynguyen2004@gmail.com (T.T.T.N.); pauserre@gmail.com (P.S.); monica.vallejop@gmail.com (M.V.-P.); morisotf@gmail.com (F.M.); 2Univ. Grenoble Alpes, CNRS, Grenoble INP (Institute of Engineering Univ. Grenoble Alpes), IMEP-LAHC, F-38000 Grenoble, France; 3Univ. Grenoble Alpes, CNRS, CEA/LETI-Minatec, Grenoble INP (Institute of Engineering Univ. Grenoble Alpes), LTM, F-38000 Grenoble, France; bassem.salem@cea.fr

**Keywords:** nanowires, nanonets, transistor, integration process, silicon

## Abstract

This paper summarizes some of the essential aspects for the fabrication of functional devices from bottom-up silicon nanowires. In a first part, the different ways of exploiting nanowires in functional devices, from single nanowires to large assemblies of nanowires such as nanonets (two-dimensional arrays of randomly oriented nanowires), are briefly reviewed. Subsequently, the main properties of nanowires are discussed followed by those of nanonets that benefit from the large numbers of nanowires involved. After describing the main techniques used for the growth of nanowires, in the context of functional device fabrication, the different techniques used for nanowire manipulation are largely presented as they constitute one of the first fundamental steps that allows the nanowire positioning necessary to start the integration process. The advantages and disadvantages of each of these manipulation techniques are discussed. Then, the main families of nanowire-based transistors are presented; their most common integration routes and the electrical performance of the resulting devices are also presented and compared in order to highlight the relevance of these different geometries. Because they can be bottlenecks, the key technological elements necessary for the integration of silicon nanowires are detailed: the sintering technique, the importance of surface and interface engineering, and the key role of silicidation for good device performance. Finally the main application areas for these silicon nanowire devices are reviewed.

## 1. Introduction

The study of nano-objects reveals a new world in which the observed properties may differ from those of bulk materials. When the size of structures reaches the nanoscale, the ratio of surface to volume of the structures increases drastically [1]. Since the number of atoms on the surface becomes comparable to the number in the volume, the physico-chemical properties of nanostructures are modified and even controlled by surface effects. These surface effects induce new properties compared to the bulk material that multiply the field of investigation. Over time, the study of nano-objects has increasingly become a scientific and technological revolution. Nanowires (NWs) can be defined as nanostructures that have diameters towards the nanometer scale (<100 nm).

Nanowires are still in an experimental stage and have not yet been used in real applications [2,3]. However, there is a graceful evolution in NW-based technology and investigations suggest that they can be used as the building blocks for the next-generation electronics and very sensitive biosensors [2,4,5]. Another possible real application of nanowires is nano-electromechanical systems (NEMS) due to their high Young’s moduli [2,6]. Nanorobots can also be aided by nanowires in the generation and conduction of their required energy [7].

In this review, we focus on synthesized silicon nanowires (bottom-up NWs) and do not consider nanowires formed by any etching techniques. These nanowires can be used vertically attached directly to their growth substrate [8,9], or detached from it to be transferred elsewhere and integrated into devices [10,11]. The mastery of the synthesis techniques allows obtaining nanowires of very small diameters, with controlled crystallinity and doping. In addition, surface engineering allows good control of their surface properties. Their quasi-one-dimensional characteristic gives them remarkable properties for many applications—such as electronic, optoelectronic, energy, and biomedical applications. In 1998, Morales and Lieber succeeded in synthesizing the first silicon and germanium nanowires with diameters of less than 20 nm for lengths greater than 1 μm [12]. This major breakthrough was an immediate success within the scientific community as a new field of research in nanoelectronics.

It is possible to make conducting (e.g., Ni, Pt, Au), semiconducting (e.g., Si, InP, GaN), and insulating (e.g., SiO_2_, TiO_2_) nanowires which can find different possible real applications [2,13,14,15]. Among these nanowires, silicon nanowires (SiNWs) are considered as popular nanomaterials due to their exceptional electrical and mechanical properties and their conductivity can be controlled by the field effect behavior [2,16,17,18]. In addition, their synthesis is very well controlled due to broad study in the literature, their integration in transistors takes benefits from the technological knowledge related to the manufacture of MOSFET, particularly for the production of electrical contacts [19,20]. Finally, NWs have a very high length-to-diameter ratio called form factor. Because of this one-dimensional shape, which is comparable to some biological molecules, NWs are the ideal transducer for bio-detection. In 2001, Lieber’s group was the first to highlight the promising potential of SiNWs as biosensors [21]. This technology still arouses curiosity today, and silicon nanowires are still potential candidates for field effect transistors and advanced sensors [22,23].

To appreciate the attractiveness of silicon nanowires (SiNWs) in a broad sense, we have shown in Figure 1 the result of a search on the Of Knowledge website with the keyword “Si or Silicon” AND “nanowire*” in the title. The first decade of the 21st century saw an explosion in the number of publications on silicon nanowires (Figure 1), as their potential appeared extremely promising. These publications are mainly concerned with the synthesis, formation, characterization, and integration into functional devices that are mostly composed of a single nanowire or several nanowires in parallel. From 2012 onwards, we observe a steady decrease in the number of publications. This can be attributed on the one hand to the less innovative aspect of the topic over the years with increasingly complete coverage of knowledge on this material. On the other hand, the difficulty of producing functional devices in a reproducible and efficient manner with technologies that are transferable to industry and at reasonable costs tends to blunt the appeal of these materials. A few years ago, and based on this observation, some groups [24,25,26,27,28,29,30] started to work on nanowire assemblies, also called nanowire networks and referred to as nanonets. Indeed, such assemblies benefit from advantageous nanometric properties, as well as easy connection to macroscopic objects thanks to their large dimension.

In this review, properties, growth, and transfer methods of SiNWs—either singularly, or in the form of a network—are reported. To have a better view of devices based on these materials, technological key parameters were discussed. Afterward, SiNW-based transistors are explored as building blocks for most of the applications depicted in different common forms—in particular, single NW field effect transistors (SiNW-FETs), nanonet field effect transistors (SiNN-FETs), and multi-parallel channel field effect transistors (MPC-FETs). Ultimately, it is shown that SiNWs, owing to their unique physical and chemical properties, are promising candidates for the wide range of applications that differ from those of bulk silicon material.

## 2. From Nanowire-Based to Nanonet-Based Silicon Devices

There are a broad variety of nanowire-based devices—such as single nanowire, crossed nanowires, forest of upright nanowires, and two- and three-dimensional (2D and 3D) nanonet based devices—as schematically summarized in Figure 2. Crossed nanowire device architectures, more complex than single NW devices, can open up new opportunities that differentiate NW-based devices from conventional paradigms. Depending on the choice of NWs, the structure can yield a variety of critical device elements, including transistors and diodes [31]. Forest-like and 3D-NW structures, combining the properties of 1D and 3D nanostructures, could have more interesting properties than simple arrays of in-plane nanowires because of their higher porosity and specific surface area [32]. However, in this review, we will focus on single NW field effect transistors (SiNW-FETs), nanonet field effect transistors (SiNN-FETs), and multi-parallel channel field effect transistors (MPC-FETs).

Today, single nanowire devices and parallel nanowire devices are well known to the scientific community and widely reported in the literature. Nanonet-based devices, on the other hand, are less well known, especially in the context of semiconductor nanowires. Nanonet, an acronym for NANOstructured NETwork, is a term that was introduced by George Grüner in 2006 [35,36]. It refers to a network of one-dimensional, randomly oriented nanostructures on the surface of a substrate. Two types of nanonets can be distinguished according to the thickness of the film formed. On one hand, three-dimensional nanonets have a thickness comparable to or larger than the length of the nanostructures (Figure 3a). On the other hand, two-dimensional nanonets are defined by a film thickness comparable to the diameter of the nanostructures (Figure 3b).

Considering the electronic properties, such three-dimensional or two-dimensional networks are governed by percolation theory. Percolation is associated with a system in which randomly distributed objects of a given geometry may or may not form connections with each other [37,38]. A nanonet is then defined as percolating when a network of infinite size can communicate (e.g., conduct current) over the entire network via percolation paths that involve nanostructures and connections between nanostructures (Figure 3b). The density of nanostructures is a key parameter to control this communication (conduction in the current, etc.) via the interconnections between nanostructures. As a result, there is a critical density, called the percolation threshold, at which percolation pathways can be used to ensure communication in the nanonet. The manufacture of functional electrical devices from nanonets requires densities above the percolation threshold. Besides, nanonets are highly interesting because once the nanostructures are gathered to form the network, new properties appear and multiply the number of degrees of freedom. These properties are separated according to the intrinsic properties of the nanostructures and the macroscopic properties that come from the nanonet itself [36,39]. The particular properties of the nanonets are described in Section 4 in detail.

The literature on silicon nanonets remains scarce, despite the very well-controlled growth of NWs and excellent carrier mobility in bulk materials [40,41]. One of the plausible reasons for the lack of interest in these nanonets is the formation of an oxide that forms around the NWs when they are exposed to air. This native oxide is an electrical insulator and thus limits conduction across the junctions between NWs and the fabrication of long channel devices [42]. As a consequence, there is a deep-seated belief in the scientific community that it would be impossible to produce functional electrical devices based on Si-NW networks. To overcome this major disadvantage, Heo et al. [42] used intermediate metal contacts in the transistor channel to allow current to flow despite insulating junctions between SiNWs, but such a solution has a profound impact on the nanonet and the advantages arising from the geometry of the nanonets are probably lost. The Lieber group [43] also made devices with arrays of silicon NWs; however, the length of the NWs remains close to the length of the channel and therefore the current can flow directly from one contact to another without using junctions between NWs. Furthermore, SiNNs have also been studied by the Unalan group for photodetection applications [44]. According to their observations, light makes it possible to reduce the height of the energy barrier at the level of the junctions between NWs, thus allowing the passage of current through the devices. However, even in that study, the length of the channel remains comparable to that of the NWs, limiting the junction number to one maximum. As a result, few NW/NW junctions are involved in the conduction of the nanonet, which prevents the full benefits of network geometry from being exploited.

Generally, semiconducting nanonets—in comparison with metallic ones—are much less studied and, most of the time, when dealing with NWs, the reported work is an isolated study that is not followed by any other publication [24,44,45,46]. We believe that the reason for this lies in the great sensitivity of semiconducting nanonets to their environment, which makes electrical properties unstable and weakly reproducible at first sight. However, Ternon’s group has demonstrated that appropriate surface engineering and optimization of the effects resulting from the nanometric scale allow addressing this instability and low reproducibility. As a consequence, sensitivity can be controlled instead of endured [47]. As a result, they developed a process for sintering junctions that makes nanonets insensitive to oxidation in the long term [48] making their integration into electronic devices possible.

Among NWs, SiNWs are considered one of the most popular one-dimensional materials due to their functionality for nanoscale electronics without the need for complex and costly fabrication facilities. The main obstacles to the mass production of single NW devices, including high-resolution lithography technologies or elaborated process technology, are still valid. There are a huge number of papers on single SiNWs, but not all the other class of SiNW assemble like silicon nanonets, although they indicate being a good alternative for single NWs due to their easier integration and production in large-scale electronics. These types of materials combine the advantage of NWs, such as high sensitivity, with the ability to be transferred on any kind of substrates, whether rigid or flexible.

## 3. Silicon Nanowire Properties

### 3.1. Mechanical Properties

Mechanical properties of nanowires are of considerable significance in device processing since changes in temperature, induced strain, and external stress can change the electrical conductivity of the nanowire due to internal dislocations or flexoelectricity [49,50]. The processing of VLSI (very large-scale integration) induces compressive and tensile stresses via deposition of different materials which can cause failure in devices mainly due to delamination and electro-migration [51]. Moreover, due to the Si NW ability to relax strain, it presents the possibility to also combine lattice mismatched materials (e.g., Ge) in axial heterostructures without the formation of misfit dislocations. Nanowires, which are 1D systems, are expected to have interesting mechanical properties due to their high aspect ratio compared to bulk materials and reduced number of defects per unit length [6,51]. However, manipulating these materials for mechanical measurements is a challenging task [51,52]. The main methods which are used to investigate their mechanical properties are mechanical resonance, atomic force microscopy (AFM), and nanoindentation [18,52]. The resonance method is used only for the determination of the elastic properties and is not easy to measure the applied force. On the other hand, nanoindentation has very good force and displacement resolution and control [52].

Experiments based on the AFM-based nanoindentation showed that the stiffness of silicon nanowires is well described by the Herz theory [18]. Therefore, the wires with diameters in the range of 100 nm to 600 nm have elastic modulus values which are independent of the wire diameter and are more or less identical to those of bulk silicon [18]. This implies that the elastic characteristics of silicon nanowires with diameters larger than 100 nm are not affected by the finite size effect. Furthermore, the elastic modulus of nanowires with diameters of less than 100 nm was found to decrease with diameter [18]. In contrast, computational studies do not support any size dependence of Young’s modulus for SiNWs with a diameter greater than 10 nm [53,54]. In fact, some experimentalists disagree on the evolution of Young’s modulus as a function of size [55]. In many cases, experimentally depicted structures and the analogous theoretically simulated models have different dimensions, surface contours, or passivation. Therefore, it has been speculated that the surface effects should have a certain influence on the difference between experimental and theoretical data. Importantly, the experimental proofs of possible sources of such misconceptions have been provided, at least for the resonance and tensile tests [56,57,58,59,60].

Moreover, NWs can be used in the field of sensors and nano-electromechanical systems (NEMS) [18,51]. This is because, according to their tensile strength and Young’s modulus, they are very robust materials and have the ability to store elastic energy [51,61]. Then, nanoscale resonators can be built from silicon nanowires with high oscillation frequencies (100 MHz up to 1 GHz) because of their excellent elastic properties [18,51]. The detection of molecules at atomic resolution was achieved with these nanoscale resonators [18].

### 3.2. Electrical Properties

The small sizes of SiNWs make their electronic and electrical properties strongly dependent on growth direction, size, morphology, and surface reconstruction. A well-known example is the size dependence of the electronic bandgap width of SiNWs irrespective of wire direction. As the nanowire diameter, d, decreases below 10 nm, the band gap of the nanowire widens and deviates from that of bulk silicon gradually (Equation (1)) [62]. Moreover, the orientation of the NW axis and the surface have a great effect on the electronic properties of SiNWs [63].
(1)$$\mathrm{Band}\;\mathrm{Gap}\propto \frac{1}{d^{n}};\;1\leq n\leq 2$$

Therefore, the electronic properties—such as the band gap, valley splitting, and effective mass—are also functions of the diameter [64]. These affect the transport properties of the nanowires [65]. Hydrogen and oxygen terminated SiNWs have also been studied to gain an understanding of their optical and electronic properties [66,67,68].

In the presence of perfect crystalline SiNW with four atoms per unit cell, three conductance channels are found corresponding to three *s* bands crossing the Fermi level [69]. Conductance variations are observed if one or two atoms are added or removed. Thus, the conductance is affected by the crystalline structure of the nanowire [23]. Furthermore, variations in the surface conditions, such as scattering phenomena of carriers in nanowires, cause changes in conductivity [23,51]; meanwhile, it is also seen that scattering phenomena are NW diameter dependent [23,70].

The large aspect ratio of nanowires makes their conductivity very sensitive to surface excitation by external charges [6,23,61]. As it is seen in Figure 4, a small surface perturbation can influence the entire section of the nanowire, whereas in the case of a thin film, only a fraction of its surface is influenced. This important property is the reason why silicon nanowires are so electrically sensitive to surface events. Thus, this phenomenon is the basis for the detection of single molecules and the use of silicon nanowires in biosensors [17,22,61,71].

It has been shown that threshold voltage (V_th_) can be significantly tuned by the NW diameter (d) [72,73]. V_th_ is linearly proportional to d in an NW field effect transistor (NWFET) due to the greater influence of the surface scattering processes. NWFETs based on thin nanowires exhibit a steep subthreshold slope with a small threshold voltage, but low conductivity in comparison with NWFETs with larger diameters in which—due to high short-channel length effects—they experience a moderated subthreshold slope with a larger threshold voltage [74].

Additionally, the free carrier concentration in silicon nanowires depends on the size of the structure. It was shown that the donor ionization energy of silicon nanowires increases with decreasing nanowire diameter. Therefore, the free carrier density can be profoundly modified at diameter values much larger (>10 nm) than those at which quantum and dopant surface segregation effects are set in [16].

Thus, silicon nanowires exhibit charge trapping behavior and transport properties, tunable by surface engineering, that make them attractive for use in electronic devices such as MIS (metal–insulator–semiconductor) structures and field effect transistors (FET). Therefore, SiNWs are attractive for designing a wide range of functional devices such as flash memory, logic devices, as well as chemical and biological sensors [51,62,75,76,77,78].

### 3.3. Surface Chemistry

As for bulk silicon, silicon nanowires are subject to surface oxidation when exposed to an oxidizing atmosphere such as air. This native oxide is of poor quality and is accompanied by many oxygen-derived defects. Moreover, native oxides are a contamination source by metallic impurities [79] and are poorly stable in aqueous media [80]. Native oxide can be easily removed by using either chemical reactants—such as buffer oxide etching (BOE) solution, diluted hydrofluoric acid (HF), HF vapor, and gaseous ammonia (NH_3_)—or by physical etching method including argon (Ar) plasma etching. Most of these methods provide hydrogenated surfaces that are an interesting starting point for further functionalization in the frame of sensor applications [79,81]. 

In contrast to bulk silicon or thin films, the oxidation kinetics of H-terminated silicon nanowires is very slowed down regardless of the oxidation temperature (ambient—900 °C) compared to planar structures [82,83,84,85]. Moreover, it has been shown that—below 100 nm in diameter—the smaller the diameter of the nanowires, the slower the phenomenon [83]. This change in kinetics is commonly explained by compressive stress at the SiO_2_/Si interface retarding the interfacial reaction [82,83]. Then, this effect becomes more significant for a more curved surface [86]. On this basis, Fazzini et al. propose that since oxidation retardation is inversely proportional to nanowire diameter, this property can be used to homogenize the diameter of a given nanowire population [87]. Such a slowing down of oxidation is an extremely interesting phenomenon when it comes to making functional devices. Indeed, this unwanted oxidation has detrimental effects on the electronic properties of nanowires. Preventing oxidation during silicon processing is an important task in all microelectronic processing. Thus, thanks to this slowing down of oxidation, silicon nanowires with a diameter lower than 10 nm can remain several days in the ambient air without oxidizing and can then be easily processed without any special precautions against oxidation [84].

### 3.4. Optical Properties

Remarkably, small diameter (<6 nm) SiNWs grown along most of the crystallographic orientations have a direct band gap [88], meaning that the maximum of the valence band and the minimum of the conduction band occur at the same point in the reciprocal space. This property has allowed envisaging the use of SiNWs as optically active materials for photonics applications [89]. Silicon nanowires exhibit strong antireflective properties and are capable of enhancing optical absorption over that of bare bulk crystalline silicon [90]. Nanowires can exhibit mechanical strain effects if exposed to light which has a wavelength comparable to their energy bandgap. This is due to their photoelastic properties [18].

In all, as yet it has been shown that SiNWs are found to process such remarkable optical properties as visible photoluminescence (PL) [91,92], very low total reflection [92,93], enhancement of Raman scattering [92,93,94], coherent anti-Stokes light scattering [95], interband PL [93,96], and efficiency of generation of third harmonics whereby light is generated at a wavelength which is one-third of the pump wavelength [92,97].

### 3.5. Thermal Properties

SiNWs could have applications in nano- and micro-scale thermoelectric power generators [98,99]. Therefore, it is important to study their thermal conductivity. However, for a major increment in the Seebeck coefficient, nanowires with a diameter of less than 5 nm are required [100]. The optimal diameter of NWs for good thermoelectric properties is between 30 and 100 nm. The thermal conductivity can be very small if rough nanowires are fabricated, while electrical conductivity and the Seebeck coefficient are very close to the bulk silicon [101]. Two opposite phenomena compete within the nanowires with respect to the thermal conductivity. On the one hand, as the diameter of the nanowires decreases, the surface-to-volume ratio increases, which increases the surface scattering effects and thus decreases the thermal conductivity of the nanowires. On the other hand, it has been predicted that for very small diameters (<1.5 nm), by a quantum confinement phonons effect, the thermal conductivity increases [102,103].

One has to note that silicon nanowires, when used within applications or experiments, may have a curved-like shape and not be straight. As the phonon transport can be affected by their curvature, their thermal conductivity then changes. There is an additional obstructive mechanism to phonon transport, particularly in the ballistic regime, thanks to the deviation of phonon from the main heat flow direction due to its curvature. Therefore, thermal conductivity is reduced when the radius of the nanowire curvature increases [5]. The effect of the curvature on the thermal impedance has a greater effect when the radius of the curvature is one order smaller than the phonon mean free path [5].

This observation is interesting since the thermal conductivity of silicon nanowires can be controlled by the proper shaping of the wire. For instance, the thermal conductivity of NWs with large roughness is found to be significantly below the prediction [5,104]. This is important in the use of silicon nanowires in next-generation electronics because the shrinking of electronic devices towards the nanoscale region demands an increase in power dissipation per unit area [99,105].

## 4. Silicon Nanonet Properties

Two-dimensional (2D) nanonets are very promising because, by an averaging effect, the structure provides an increase in the reproducibility of nanonet properties by minimizing the disparities existing between the NWs. It also compensates for potential failures in operation to the extent that NWs from the network, which are not initially involved in conduction paths, can contribute to a new conduction path to bypass the failed NWs. Nanonets offer undisputed advantages thanks to the wise combination of the intrinsic properties of nanostructures with those of the nanonet. According to Zhao and Grüner, the ‘nanonet’ morphology can be regarded as a fourth material phase in addition to monocrystalline, polycrystalline, and amorphous structures [106]. Finally, because of the coexistence of NWs and junctions within a conduction path, it is expected that the switching of the transistors from ‘off’ state to the ‘on’ state will be even more abrupt as the number of junctions increases [107]. However, to allow these advantages to be revealed and exploited, it is important that the dimensions of the nanonet be much larger than the length of the NWs and that the number of NWs involved in conduction is sufficiently large. Here, important parameters impacting nanonet-based devices besides ones from individual NWs are the morphological quality of the nanonets, the density of NWs within the nanonet, the size of the nanonet, and particularly the distance between the two electrodes of the device compared to the length of NWs. Such degrees of freedom in nanonet create vast opportunities in future applications.

### 4.1. Electrical Conductivity

Above the percolation threshold, the current is likely to flow through percolation paths (also called conduction paths) that involve nanostructures and connections between nanostructures. These junctions between nanostructures appear as an energy barrier for carriers and are therefore likely to be more resistive than nanostructures [36,108]. In that case, they can control conduction within the nanonet. As mentioned earlier, these junctions generate additional resistance, so they can enormously affect percolation transport in nanowire network [107,109]. Such a two-dimensional network is normally governed by the theory of percolation. Percolation is often defined as a system in which randomly distributed objects of a given geometry can form, or not form, connections between each other [37,110,111]. A nanonet is then defined as a percolating system when a network of infinite size can communicate (e.g., conduct current) over the entire network via percolation paths that involve nanostructures and connections between nanostructures. The density of nanostructures is a key parameter to control the communication in the medium via the interconnections between nanostructures. As a result, there is a critical density—called the percolation threshold—above which percolation paths allow for communication in the nanonet. The fabrication of functional electrical devices from nanonets requires the use of densities above the percolation threshold.

### 4.2. Porosity and Optical Transparency

Due to the very high aspect ratio of nanostructures, and by playing with the density and diameter of NWs, NNs can reach even 90% of optical transparency as illustrated in Figure 5.

Indeed, for low-density percolating networks, nanonets are essentially composed of voids. In the case of metallic NWs, this property makes nanonets particularly attractive as transparent electrodes [26]. With the same idea, in the case of semiconductor NWs, nanonets would be particularly attractive to form transparent transistors and transparent electronics in general. Moreover, the high porosity of the nanonet can allow the insertion of functional materials [108], which is of high value when dealing with biosensors.

### 4.3. Mechanical Strength and Flexibility

Based on the excellent flexibility in individual nanowires which is diameter dependent, and for diameters less than 100 nm is remarkable when these nanostructures are assembled as a nanonet, the entire network is capable of being subjected to mechanical deformation [112] (Figure 5) and can adapt to the substrate morphology [112,113]. The flexibility in the choice of nanostructures and the unique structure of the network suggests a broad spectrum of applications for nanonets, as we will see in the following.

### 4.4. Fault Tolerance and Reproducibility

For electrical devices, conduction in a nanonet is ensured by multiple percolation paths that connect the two metallic contacts. Then, the number of conduction paths is greater but the presence of junctions within each path ultimately implies a reduction in current. Therefore, for a given electric field, the amount of current is generally lower in nanonet than in single nanowire devices (Figure 6). However, if one path is faulty, many other conduction paths remain and can guarantee the functionality of the device [36].

Moreover, the macroscopic properties of a given nanonet are the result of a considerable number (for 1 × 1 cm^2^ e.g., 10^6^–10^8^ NWs) of nanostructures which makes it possible to smooth out the disparities that may exist from one NW to another. Therefore, the properties of the nanonets show less disparity than those of the population of nanowires that constitute it. This can be demonstrated by studying the electrical characteristics of the single nanowire or nanonet devices based on the same nanowire population, as Pauline Serre has done. With a technological work based on heavily doped NWs (degenerated), she produced single SiNW resistors and Si NN resistors (Figure 6) [28,39]. By studying numerous devices (10 single SiNW resistors, >100 SiNN resistors), evidence of averaging effect is proven for nanonets. Indeed, the dispersion in the current of these two types of devices is clearly different: 64% for single NW resistors, against 18% for SiNN resistors. As a consequence, one of the main drawbacks of the single NW-based devices, lack of reproducibility, can be eliminated by the nanonet geometry.

## 5. Silicon Nanowire Growth

There are two major approaches to form nanowires. The ‘top-down’ approach starts from bulk silicon in the form of a substrate or thin film silicon with the use of SOI substrate with the objective to etch the material until reaching the formation of high aspect ratio structures called nanowires, but which section is not necessarily circular. Besides, the ‘bottom-up’ approach is used to grow nanowires, either vertically on the substrate or in-plane lying on the substrate. In the past few decades, there has been extensive research on synthetic nanowire strategies focused on a bottom-up approach to understand the growth of nanostructures, tune the geometrical dimensions during growth, and form heterostructures. Here, we focus on bottom-up approaches that give the opportunity to detach the NWs and to collect them with the objective to form a nanonet by assembling the NWs. Then, targeted bottom-up syntheses are additive and can be done in two phases: vapor and solution phase.

### 5.1. Vapor Phase

#### 5.1.1. Low Pressure

In the frame of low-pressure chemical growth, silicon nanowires are synthesized using the vapor–liquid–solid (VLS) mechanism (see Figure 7A) on a silicon substrate <111> from a metal catalyst (Au [115,116], In [117], Pt [118], Sn [119], etc.) in a CVD reactor. To allow the NW growth, the catalyst must be present on the surface of the substrate in the form of nanoparticles. For this purpose, it is possible to disperse a solution of colloids on the surface of the substrate [11] or to deposit a thin film that forms nanoparticles by thermal dewetting [117]. Then the growth of the Si nanowires is carried out in the presence of silane, SiH_4_, used as a precursor to silicon. The Si atoms incorporate into the metallic droplets and when it reaches saturation, the excess silicon crystallizes at the interface between the droplet and the substrate. This process continues as long as the system is supplied with gaseous precursors and thus forms silicon nanowires.

Silicon nanowires can be doped during growth by adding boron (p-doping) or phosphorus (n-doping) in gaseous form. Phosphine, PH_3_, is used for n-type doping and diborane, B_2_H_6_, for p-type doping [120]. The concentration of dopants in nanowires is directly related to the ratio between the concentration of dopant gas and the concentration of silane [121], which allows the doping of nanowires to be controlled with great precision during the growth. Two other gases can also be added during the synthesis of nanowires: hydrogen, H_2_, which is used as a carrier gas; and hydrogen chloride, HCl, which is used to prevent the diffusion of gold on the surface of the nanowires and thus inhibit the lateral and branch growth of the structures [121,122,123]. The silicon nanowires produced by VLS are of good crystal quality and can have a very high length at low costs. The length of the nanowires is controlled by the time of the growth and diameter of the nanowires depending on the size of the metallic nanoparticles [21].

#### 5.1.2. High Pressure

The patented SyMMES technique [124] is similar to the CVD method and it aims to make its industrial scaling simpler, by limiting the hazard of the reactants, facilitating reactor design, and producing nanowires in large quantities.

In Figure 8, this method is shown for the solvent-free chemical synthesis of thin (10 ± 3 nm) SiNWs using diphenylsilane as a Si source and small (1–2 nm) gold nanoparticles (AuNP) as a catalyst in a sealed reactor at 420 °C and with a pressure < 10 bar. The catalyst nanoparticles are immobilized on micron-sized salt (NaCl) powder, which acts as a sacrificial 3D substrate which is easily removable by washing with water after NW growth. Pure SiNWs are obtained at a high production yield of 1 mg cm^−3^ of reactor volume and with a 70% chemical yield. In this method, n-type doping of the SiNWs is achieved by adding diphenylphosphine at concentrations of 0.025 to 1.5% as the dopant source [125]. Recently, the impact of the size/shape of SiNWs grown by this technique is also demonstrated on the electrochemical performance of conventional Li-ion batteries [126].

### 5.2. Liquid (Solution) Phase

The first liquid phase technique for SiNW growth is the direct counterpart of the VLS technique. Then the solution–liquid–solid (SLS) mechanism is fully similar to that of VLS, except that nanowire precursors are dissolved into a high-boiling liquid, such as squalene (C_30_H_62_), and the catalysts are suspended therein, as described by Heitsch et al. [127].

The second method in liquid involves electrochemical phenomena. Here, anodic aluminum oxide (AAO) substrates are used for templated solution growth, using electrochemical deposition to fill the channels as shown in Figure 7B [128]. Drawing upon the solution-phase synthesis of nanoparticles, redox reactions can also be used to produce nanowires [129]. Seed particles are first grown by a rapid reduction of a dissolved precursor with a strong reducing agent such as sodium borohydride. Secondary growth is achieved with a weaker reducing agent, such as L-ascorbic acid, to prevent additional seed particle production. The nanowire anisotropy is achieved by controlling surface chemistry [123].

### 5.3. Summary on Growth

Table 1 gives a simple comparison between the different techniques used for SiNW growth. We studied the various methods currently used in bottom-up SiNW growth. Gold is the most widely used catalyst for SiNW growth by CVD under VLS mechanism, as it offers a good size control. However, the recent advances in the mentioned SiNW growth techniques are still in their early stages. The most appropriate method for the growth will ultimately be determined by a number of factors, including the desired application, as well as the available process control and associated costs. SiNW properties can be finely tuned for the desired application using the various growth methods available.

## 6. SiNW Collective Handling

After growth, the nanowires are detached from the substrate and dispersed in solution to allow collective manipulation (colloidal suspension). Depending on the chosen assembly technique, it is possible to form arrays with a preferred orientation which are then not necessarily percolating or to form randomly oriented arrays. The common points of all these methods are the low thermal budget (less than 200 °C) and the easy scalability. In the frame of electronic application and functional device formation, the formation of the percolating network is mandatory. As a consequence, when the applied method leads to a preferential orientation, a two-step process has to be used to allow the crossing of the NWs that is required to allow charge carrier displacement.

### 6.1. Network with Preferential Orientation

#### 6.1.1. Drop-Casting

The drop-casting is the simplest method to deposit NWs on top of a substrate. In this approach, a drop of a NW suspension is first deposited (literally ‘drop-casted’) on the surface. Second, a drying process is applied to evaporate the liquid. The clear advantage of this approach is its simplicity and versatility. There is also an opportunity to even mix different nanowires in a suitable solvent via ultrasonication before drop-casting. The main drawback is related to the homogeneity of the NW distribution on the substrate surface. Indeed, the drying process often induces shear forces that impact the NW positioning. As a consequence, drop-casting is well dedicated when the needs in terms of positioning or density are not too restrictive.

#### 6.1.2. Fluidic Directed Assembly

In this method, nanowires can be assembled into parallel arrays with the control of average separation as well as complex crossed nanowires arrays. In this technique, the nanowires are suspended in a solution such as ethanol. Then, the suspension is sent through fluidic channel structures formed between poly(dimethylsiloxane) (PDMS) molds and a flat substrate. As a consequence, the NWs tend to align in the flow direction. In this way, by building the appropriate fluidic channel, parallel and crossed arrays of NWs can be readily achieved with single and sequential crossed flows (Figure 9), respectively, for the assembly [130]. The same result can be obtained with chemically patterned substrate that preferentially attracts the NWs in the functionalized region (Figure 9A).

The fluidic phase poses a challenge for single-nanowire control, accompanied to some extent by issues related to spacious microfluidic and large footprint electrodes [130].

#### 6.1.3. Langmuir–Blodgett Assembly

Another promising method to produce aligned arrays of NWs on various substrates is the Langmuir–Blodgett (LB) technique [43,131,132,133,134]. In this technique, NW suspension is densely packed using a compression trough at an air–liquid interface. Dip coating can be then used to transfer the NWs onto receiver substrates by van der Waals, hydrophobic-hydrophilic, or electrostatic interactions when the substrate is lowered and withdrawn from the system, in the vertical direction as shown in Figure 10a.

This technique is frequently used in the assembly of highly ordered nanomaterials. By repeating the assembly process with a changed orientation of the substrate, hierarchical nanowire structures can also be produced as shown in Figure 10b–d [131].

The density of nanowires can be improved by adapting LB techniques to align the nanowires. Limitations of this technique include reorganization of the nanowires during dip coating that leads to overlapping features and gaps within the dense arrays of nanowires [43,133,134].

#### 6.1.4. Blown-Bubble Films

An approach with the potential for large scale transfer of well aligned NWs is the so-called blown-bubble film assembly (BBF) (Figure 11). This approach involves the preparation of homogeneous polymer suspension of NWs, followed by the expansion of polymer suspension using a circular die, and finally the breaking by an external force and the transfer of the bubble film to the desired substrate. By tuning the preliminary concentration, well aligned, and controlled density NWs over large areas are achievable [135,136,137].

Blow-bubble approach is inexpensive and can be adapted to patterning nanowires on many different types of substrates (e.g., flexible, flat, or curved). However, a challenge for these techniques is to control the viscosity of the bubble film in balance with a compatible surface coating on the nanowires, which has so far limited this demonstration to films of epoxy. Another challenge for this technique and for the alignment achieved using microfluidic flow is to precisely control the position of the deposited nanowires [43,136,137]. This method is highly attractive in a technology where device fabrication costs must be kept to a minimum and for transferring and aligning a variety of nanomaterials including SiNWs and CNTs [136,137].

#### 6.1.5. Contact Printing

‘Printing’ has usually been used to describe a method by which a layer of ink is transferred from a stamp to a substrate through a reversing reaction [138,139]. Figure 12 contains an illustration of two different printing apparatuses. Printing methods thus include flexographic printing, offset printing, gravure printing, screen printing, and ink-jet printing. In the printing method, the choice of the solvents for ink preparation has been identified as an important parameter for active layer surface morphology [139,140,141,142].

A simple but promising method is the contact printing (CP) technique developed by Javey et al. Here, the particularity is that the NWs are kept on their growth substrate until the transfer, no need to form a suspension. Therefore, during CP, nanowires are mechanically transferred by a shearing motion between the growth and the target substrate. In order to maximize the nanowire density and the alignment yield, the use of lubricants—such as mineral oil—is essential [143].

Yao et al. have significantly developed the nanowire alignment in CP by nanoscale combing technique [144,145]. In this technique, patterned resist windows opened by lithography are utilized to store the nanowires partially within the so-called anchoring regions [143,144,146], see Figure 13.

Inspired by this CP technique, Robkopf and Strehle expand the current scope by focusing on the low-density nanowire assemblies and on dry friction to confine the nanowire deposition during CP [143]. Their motivation for dry friction was based on the fact that lubricants might act as a source of contamination in micro-device fabrication [147]. Therefore, they developed the concept of surface-controlled contact printing (SCCP) to avoid the need for lubricants, patterned resist, and post-print resist removal procedures. SCCP method is based on the frictional force between an individual nanowire on the growth substrate and the target surface, as shown in Figure 14. It has been shown that the material of the surface, the surface roughness, elevated structures, and nanoparticles can, in principle, be effective in the positioning of nanowires.

Contact printing can be a promising alternative to transfer highly controllable single-NWs, but using lubricant is still necessary for that. SCCP provides further possibilities to control the positioning of nanowires by lubricant free or dry friction interaction. The reduction in nanowire densities and lower alignment yields are still some drawbacks of this method [143,144].

### 6.2. Random Networks

#### 6.2.1. Vacuum Filtration

Vacuum filtration [148,149] is widely used in the literature as it allows the production of homogeneous nanostructured networks at low cost and over large areas. During solution filtration, the nanostructures are randomly trapped on the surface of a porous filter. In the frame of Si nanonet assembling, Pauline Serre [28] and Maxime Legallais [27] have shown that the process consists in five main steps that are: (1) dispersion of the silicon NWs in solution, (2) purification of the suspension by centrifugation, (3) analysis of the suspension by absorption spectroscopy, (4) assembly of the NWs into nanonets by vacuum filtration, and finally, (5) transfer of the nanonet onto a substrate, as shown in Figure 15.

Since the properties of nanonets are influenced by NW density, it is important to be able to control the density of nanonet. As the density of nanowires in nanonets is directly related to the number of nanowires in the suspension, the mastery of a technique of quantification of NWs is necessary. In Ternon’s group, several techniques—such as Raman, infrared, fluorescence, and absorption spectroscopy—have been tested. Only the latter has made it possible to obtain a measurement that is directly proportional to the number of nanowires in the solution [28,36,150]. Then, for a given absorbance and a given NW geometry, the NN density is directly linked to the volume of the filtered suspension with high reproducibility. The greater the volume of filtered suspension, the greater the density in the nanonets (see Figure 16).

Once the nanowire solution has been analyzed, the silicon nanowires can be assembled by the filtration method with the schematic equipment in Figure 15 (step 4) [28,151,152]. The nanowire suspension is filtered for a few minutes through a nitrocellulose membrane. The progressive accumulation of the nanostructures on the filter surface decreases the flow velocity in these areas and induces an increase in flow in the areas devoid of nanostructures. These different flow velocities involved are at the origin of the self-assembly mechanism and the homogeneity of the nanonet.

When the nanowire suspension is not homogeneous—i.e., it contains aggregates from growth defects or clusters of NWs—it is possible to remove some of these elements by centrifugation as demonstrated by M. Legallais [27].

After filtration, NN can be transferred to the desired substrate either by the dissolution of the filter [148] or by direct contact [153]. For instance, Serre et al. [28] and Legallais et al. [27] demonstrated that this transfer can be carried out via a wet process by dissolving the membrane produced by vacuum filtration in an acetone bath. The adhesion of the nanonets to the surface of the substrates is simply due to van der Waals forces [27,28]. This process is well dedicated for SiNN [44,154] and can be adapted for a wide variety of nanostructures—such as NWs of zinc oxide [155], germanium [156], or carbon nanotubes (CNTs) [148,152]. Furthermore, the size of the nanonet is only conditioned by the size of the filter used and can therefore be easily enlarged. Finally, the formed nanonets can be transferred onto different types of substrates either rigid or flexible, insulating or conductive, opaque or transparent, as shown in Figure 5, depending on the characterization and the intended application. Still, there is some difficulty such as finding a flexible substrate which is compatible with acetone.

Vacuum filtration is a low-temperature process in film production which affords films with some advantages such as surface uniformity and controllable thickness. Transferring NWs to a flexible substrate after being deposited on the filter depends on the substrate endurance towards acetone as most of the flexible substrates are attacked by it [27,157].

#### 6.2.2. Spray Coating

One of the simple and efficient routes for the deposition of randomly dispersed nanowires or highly ordered and highly aligned even on a wide range of receiver substrates is spray coating. Spray coating is a technique in which nanowire suspension is electrostatically forced through a nozzle whereby a fine aerosol will be formed [158,159]. The spray coating system consisted of a hot plate for controlling the temperature of the substrate, a pressure flow spray nozzle element, a nozzle movement, and an angle control module. Normally, the spray coating system is designed by taking to account the viscosity of NW suspension, the NW suspension supply, and other process variables [158] (see Figure 17). Ossama Assad et al. [158] showed that by controlling these conditions and provided that the size of the generated droplet is comparable to the length of the single NW, the shear-driven elongation of the droplet results is likely in the alignment with the confined NW in the spraying direction.

Since this technique has no limitation in substrate size, it has great potential for large scale production and can replace spin-coating which is a conventional method [139]. However, the main concerns of utilizing the spray coating belong to higher film thickness and roughness.

As already mentioned, one of the key parameters in nanonet assembly is density. In spray coating, by controlling the concentration of the nanowire suspension or at the same time regulating the flow duration, it is possible to adjust the density of deposited NWs—e.g., for having high density deposited NWs, we should increase the spray coating duration. The level of density control over deposited NWs in this approach is similar to other techniques such as blown-bubble and contact transfer techniques, while in these two techniques we are dealing with just aligned nanowires.

Spray coating facilitates a low-cost uniform coverage over a large area which has a potential for immediate implementation in the industry and/or line production. In this technique, NWs film making by the spray coating method depends on the nozzle speed, diameter, and length of nanowires and is more beneficial for high densities and large-scale fabrication [158,159].

### 6.3. Advantages and Disadvantages of Each Technique

As already discussed, semiconductor NWs have demonstrated excellent performance for nanoscale electronics and due to their great mechanical flexibility, high yield, and low-cost bottom-up synthesis; they have outstanding potential to be used in flexible electronics. As the assembly of NWs remains a challenge for practical large-scale application, various innovative NW assembly technologies have been investigated [160]. As a result below in Table 2, there is a simple comparison between different techniques of transferring nanowires into a variety of substrates from rigid to flexible ones.

In order to fabricate devices based on a single SiNW, the best option in terms of simplicity and adaptability to varieties of substrates is drop-casting, but still this technique is not efficient and the number of devices out of each NW transfer is trivial and negligible. However, other techniques—such as Langmuir–Blodgett, blown-bubble, contact printing, and fluidic directed—are the best choices for multiple-parallel channel devices. Each technique has its advantages but still, Langmuir–Blodgett seems to be a better option over others. Finally, in order to have nanonet (assembly of randomly oriented nanowires), we deal with mostly two methods, vacuum filtration and spray coating in the other technique several transfer process in different angles are needed to produce arrays of nanowires with junctions so these latter methods are complex with low yield. Although in vacuum filtration technique, density regulation is much easier and the process is inexpensive, spray coating can prevail over vacuum filtration because it is easily applicable on different substrates, at large scale, and it allows the selection of a random or aligned nanowire array.

Generally, though all these methods have an advantage over direct growth methods, these methods extend the assembly procedures, which increase the risk of additional contamination and destruction of the intrinsic properties.

## 7. Silicon Nanowire-Based Transistors

The small size and unprecedented ability to combine semiconductors with very different lattice parameters provide exciting new opportunities for devices. At the beginning of the 20th century, nanowire device researchers face the exciting challenge of deciding which devices, and thus which future applications, hold particular promise for this new class of material [161,162]. Among the many possibilities, the field effect transistor (FET) stands out as the modern workhorse of the semiconductor industry. Not surprisingly, most of the efforts in nanowire devices have focused on the fabrication of nanowire field effect transistors, as it is the building block of modern electronics and the most frequently fabricated device in history. FET is the dominant semiconductor device in digital and analog integrated circuits (ICs), and the most common power device [163]. It is a compact transistor that has been miniaturized and mass-produced for a wide range of applications.

Silicon transistor technology, especially metal–oxide–semiconductor (MOS) technology, is scaling down as predicted by Moore’s Law [18,164]. Originally, transistors had a three-dimensional active region (solid silicon substrate) then it was reduced to a two-dimensional geometry (silicon ultra-thin film on insulator (SOI)) to finally reach a one-dimensional structure with the introduction of nanowires (FinFET on SOI) [165,166], as schematically illustrated in Figure 18.

However, this scaling of the typical silicon transistor technology has almost reached its limits [22,51]. Despite the advanced capabilities of fabrication tools, many physical effects can prevent the tools from performing satisfactorily when the device size is scaled down to the nanoscale. In addition, transistor scalability, performance, and power dissipation are three fundamental issues facing aggressive miniaturization. Furthermore, the leakage mechanisms associated with the size reduction of conventional silicon transistors down to the nanoscale include direct gate dielectric tunneling, band-to-band tunneling, and short channel effects. Finally, silicon has almost reached its intrinsic switching speed limit. Other semiconducting materials with higher carrier mobility could solve the problem, but the required power dissipation levels in the nanoscale structure do not allow further downscaling [6].

Nevertheless, one-dimensional (1D) structures are the smallest structures that can be used for efficient transport of electrons [6,61,167]. Semiconductor nanowires fall into this category and are obvious candidates to replace ultrathin-film SOI transistors [168]. Among semiconductor NWs, silicon nanowires have been studied more and are considered the most suitable for implementing nanowire transistors, as silicon dominates the semiconductor industry and its structure and doping can be precisely controlled [168]. However, the smaller the footprint of the manufactured devices, the higher the manufacturing cost and the more complicated it is to observe good reproducibility and homogeneity of the device characteristics. Nevertheless, being able to introduce nanometric structures into the devices is extremely interesting because it allows new properties and functionalities to emerge. In this, nanonet devices are extremely promising because they are made of nano components while having a large enough footprint to lower the cost and complexity of production. They are clearly not part of the miniaturization dynamics of the ‘More Moore’ trend, but rather contribute to the ‘More-than-Moore’ trend or even to a paradigm shift (Figure 19a).

Nanonet based transistors belong to the category of thin film transistors that provide numerous functionalities for various applications such as CMOS circuits, sensors, responsive surfaces, and flexible and transparent electronics. They compete not only with other materials such as oxides or organics but also with other forms of silicon, amorphous, polycrystalline, or monocrystalline. In this context, Figure 19b shows the performance of these devices according to their footprint.

In this section, we focus on nanowire transistors with the simplest geometry—namely, a source electrode, a drain electrode, a channel consisting of a single nanowire, parallel nanowires, or a nanonet, as well as a full back gate or localized top-gate. Thus, we will present the most basic integration processes and then the typical electrical characteristics.

### 7.1. Integration Process

#### 7.1.1. Single Nanowire FETs (Single-SiNW-FETs)

Whether the approach is bottom-up or top-down, there is a phenomenal variety of integration processes and device geometries in the literature. There is a large number of studies presenting the fabrication of transistors with the simplest geometry, which we will describe in the rest of this paragraph [10,11,169], to the most complex transistors with variable doping along the nanowires and gate multiplicity allowing an electrostatic control of the doping and thus the control of the nature of the channel (P or N) [170,171]. Even for the simplest geometry, there are integration subtleties that impact the properties of the manufactured transistor. A simple silicon nanowire transistor can be built as shown in Figure 20a. The two gold cubes represent the source (labeled S) and drain (labeled D) contacts of the transistor respectively whereas the gate is on the backside of the substrate. The dark blue cylinder represents the nanowire which is the channel of the transistor. The channel can be doped p-type or n-type. The top-gate of the transistor is shown in Figure 20b,c as a rectangular-like plate labeled with G. The gate can be placed in a semi-cylindrical shape on the top of the nanowire or all around the nanowire [168]. SiNW transistors manufactured with a gate that surrounds the whole nanowire (all-around-gate) allow better current control through the channel and thus higher current densities can be controlled compared to a planar device. What is more, the all-around-gate allows the devices to be shrunk even more (down to 10 nm) due to the excellent control of short-channel effects and leakage [51].

The bottom-up approach, which is the subject of this paper, has caught our attention because it provides great flexibility in that nanowire growth and integration are two completely independent steps. Thus, it is easy to choose the material, length, diameter, type, and concentration of the dopants, and the orientation of the crystals because all these parameters can be adjusted during the synthesis. Today, the main limitation of this approach is the complexity of developing a method for large-scale transistor integration with high reproducibility.

For the simplest geometry (Figure 20a), the bottom-up processes start with as-grown SiNWs in both bottom-up (Figure 21i) as described in Section 5. Subsequently, the as-synthesized SiNWs have to be transferred onto the desired substrate, generally including the back-gate, by using the simple drop-casting or one of the methods described in Section 6 (Figure 21ii). In some works, SiNWs are passivated in order to preserve them from the environment sensibility, low efficiency, and short lifetime [27,172,173]. Thereafter, a photo- or electron-beam lithography process (Figure 21iii) followed by metallization (Figure 21iv) and lift-off (Figure 21v) are utilized to pattern metallic electrodes to the SiNWs [3,169,174]. Silicidation which is a process of the intermetallic compound formed by the reaction of metal and silicon can be performed in order to lower the access and/or series resistance of the device in contacts [11,175,176]. Eventually if desired, a top-gate can be added by an additional photolithography step.

Generally, this type of integration process, based on drop-casting, yields only a low success rate in obtaining nanowire devices without allowing large statistical studies. However, this is sufficient to study the fundamental properties of single SiNW devices. Furthermore, this method is clearly not suitable for the low-cost, mass production of SiNW devices. Therefore, the use of the techniques described in Section 6 that allow the assembly of SiNWs with controlled orientation and spacing over large areas is essential for the fabrication of complex logic circuits and high-performance devices.

#### 7.1.2. Silicon Nanonet FETs (SiNN-FETs)

To qualify as a nanonet device, the geometry of the transistor must be such that the length of the channel is greater than the length of the nanowires so that no nanowires are able to bridge the two electrodes directly. The current flows from one contact (source) to another (drain) via percolation paths that involve NWs as well as junctions between NWs. Typical devices are illustrated in Figure 22.

Thus, the fabrication starts with the assembly of the nanonets on the surface of the substrate including the back gate according to one of the methods described in Section 6. Then the fabrication process of back-gated SiNN-FETs can be broken down into four main steps:-Sintering of the NW–NW junctions and passivation of the NN to stabilize electrical properties (see Section 8.1 Sintering and Section 8.2 Surface and Interface).-NN patterning to define the channel geometry (Figure 23i-0–i-5).-Deposition of the source/drain contacts (Figure 23ii-0–ii-5).-Silicidation of the source/drain contacts (see Section 8.3 Silicidation).

By doing the silicidation step, back-gate SiNN-based FETs are achieved. These steps can be modified due to the need and application. For example, one might need a local top-gate instead of a full back-gate hence some procedures will be added such as deposition of the gate oxide, lithography of top-gate, and metal evaporation step to introduce top-gate. In Figure 24, SiNN-based FETs with two different designs are demonstrated.

#### 7.1.3. Multiple-Parallel-Channel FETs (MPC-FETs)

The realization of a multiple-parallel-channel field effect transistor (in reference to multiple parallel NWs as channels) can arise from two strategies. First, by choosing an appropriate technique (Section 6), it is possible to align nanowires parallel to each other and then apply the integration process presented for single nanowire transistors which will then result in an MPC FET since the channel will consist of nanowires in parallel (Figure 25a,b). The second strategy consists in producing a nanonet on the surface of the substrate and then applying the dedicated integration technique while choosing a geometry such that the length of the channel is smaller than the length of the NWs. As a consequence, several NWs are able to bridge the two electrodes, forming an MPC-FET (Figure 25c,d). 

### 7.2. Electrical Characteristics of Single SiNW-, SiMPC-, and SiNN-FETs

The operation of a single SiNW-based transistor is similar to that of a typical FET transistor. If the SiNWs are p-type and a positive/negative voltage is applied to the gate (G), then the carriers are depleted/accumulated; conversely, if the SiNWs are n-type and positive/negative voltage is applied to the gate, then the carriers are accumulated/depleted (Figure 26). Therefore, the variation of the SiNW conductance via the field-effect action allows for the transistor action to be implemented with SiNWs [17,31].

P-type SiNWs have attracted greater interest than n-type ones. P-type SiNWs were fabricated and showed high performance characteristics. Their transconductance was about 10 times greater than typical planar devices and the holes’ mobility was an order of magnitude larger too. Both the transconductance and mobility increased after surface modification of the SiNW. This implies that some of the electrical characteristics of the SiNW transistors can be controlled by proper surface modification as described in the next section. Therefore, the performance of SiNW transistors can exceed that of typical devices [51].

In addition, one of the major gains in 1D nanostructures is depletion or accumulation in the bulk of nanowires while in 2D structures such as thin film is happening only on the surface which causes lateral current shunting. This property provides sensing with label-free and direct detection when the nanowire is used as a sensor to make real-time detection possible [22,61]. Based on this advantage in nanowires, Li et al. [75] depicted an efficient strategy through surface functionalization to build a single silicon nanowire field-effect transistor-based biosensor that is capable of directly detecting protein adsorption/desorption at the single-event level (see Figure 27).

As will be discussed in the following section, the contact resistance of SiNW-FETs affects their performance significantly. Devices with titanium (Ti) source-drain contacts revealed variations in their transconductance and mobility after thermal annealing effects. Other materials for the contacts, such as silicide and nickel monosilicide, can reduce this variation in the performance of the device [11,51].

That is why the MPC-FETs and NN-FETs are particularly interesting, as they can maintain the nanosize of the components while increasing the amount of current for a given voltage [178].

In the case of MPC-FETs, fabricated from aligned parallel NWs, Figure 28 depicts the off-current as a function of the on-current for different inter-electrode (IE) spacing (2.5–5.5 μm). It can be seen that the current output can clearly exceed the µA range and that it increases when the inter-electrode (IE) spacing is reduced. However, the off-current increases when the on-current increases, and accordingly, the on/off current ratio drops with decreasing IE spacing. Since channel length (*L_c_*) of individual nanowires varies in the parallel array due to dispersion in nanowire alignment as shown in Figure 28b, though a small number of nanowires with short channels will be found in the ensemble, affected by quasi short channel effect (SCE), these nanowires will degrade the total on/off current ratio. This apparent quasi short channel effect for parallel array devices at such a large electrode spacing is not observed for single nanowires [178].

When MPC-FETs are obtained from a randomly oriented nanowire network, the same trend is observed: by decreasing the channel length, the on and off current increase (Figure 29A,B, 5 and 10 µm). Moreover, from this figure, one can notice a surprising and significant degradation of the subthreshold swing.

Then MPC-FETs are interesting to increase the on-current, but at the same time, off-current also increases and the commutation ability degrades. Such behavior can be explained by the intrinsic disparity of the nanowire population. Indeed, whatever the method of bottom-up synthesis of NWs, it is possible to observe variations of diameters, doping, or surface properties from one nanowire to another. Such variations are responsible for different electrical behaviors for each NW. Thus, there is a dispersion of on-current, off-current, sub-threshold slope, and threshold voltages [179,180]. While this is not a major problem for the on-current, the existence of a single nanowire with a high off current will result in a high off current for the MPC FET. Similarly, if all of the nanowires making up the MPC-FET switch successively, this will result in the MPC FET switching over a wide range of gate voltages [72].

In this context, NN-FETs appear to be promising for reconciling high current, high I_on_/I_off_ ratio, and fast switching, as illustrated in Figure 29 (15–100 µm). In Figure 29B, by measuring over 70 NN-based transistors with different channel lengths, an interesting trend in the I_on_/I_off_ curve is observed. On one hand, the NN-FETs with the smallest length (15–20 µm) have a quite dispersed current while the current is almost stable. On the other hand, for the longer (>20 µm), current is almost fixed with a reduction in current by increasing the channel length. As a consequence, this leads to a drastic change in the on-to-off ratio and the observation of an optimum on-to-off ratio as high as 10^5^ [72].

In order to have a better understanding of the differences inherent to these three types of nanowire-based devices, Figure 30 presents the transfer characteristics of a single-SiNW FET, an MPC FET, and an NN FET. All previous observations are summarized on that graph. The MPC-FET clearly exhibits the highest on-current, but all other transistor parameters are poor. In contrast, and against all expectations, the NN FET maintains very good performance with fast switching and very low off-current while exhibiting an on-current up to 100 times higher than that of the single nanowire device, even though its channel is over 10 times longer. As a consequence, an NN-FET could be as interesting, and even more than a single SiNW-FET.

The statistical summary in Figure 31 presents a comparison between the performance, in terms of I_on_/I_off_ ratio and subthreshold slope, of single SiNW FETs from the literature and SiNN as a function of channel length. It is important to mention that, for SiNN based transistors, each point on the graph results from measuring the performances of several devices (average and variability): on the order of 15 devices for channel lengths below 300 µm and 3 to 6 devices for longer ones.

Such a graph clearly shows that single-nanowire devices exhibit highly dispersed performance, even for a given channel length, while NN devices show much lower dispersion.

In conclusion, despite having lots of resistive junctions, SiNN based devices have good performance. Then the NW/NW junctions, which might initially appear to be a weakness of this type of system, are in fact an asset that allow good performance, good reproducibility, and high fault tolerance.

## 8. Technological Key Elements for SiNW-Based Device Integration

### 8.1. Sintering

When silicon is exposed to air, silicon dioxide (SiO_2_)—commonly known as native oxide—is systematically formed on its surface. This oxidation is self-limiting at ambient temperature and the SiO_2_ formed stops growing once it has reached a certain thickness of a few nanometers. According to high-resolution transmission electron microscopy (HRTEM) analyses, the NWs are covered with about 2 nm native oxide [28]. With a permittivity dielectric of 3.9, the silica layer—of amorphous structure—has good electrical insulating properties. The native thickness formed around the NWs is sufficient to inhibit the possible conduction by tunneling from one NW to another and makes it impossible to fabricate percolating devices with SiNNs [42]. Subsequently, Pauline Serre showed that in the absence of oxide, current can flow from one nanowire to the other via the junctions [39]. To do this, she studied the behavior of resistors based on NNs. Thus, as long as the devices are stored under nitrogen after deoxidation by hydrofluoric acid (HF) treatment, significant conduction is observed across the resistor. As soon as they are exposed to air, conductance decreases exponentially with air exposure time due to the progressive reoxidation of the NWs as shown in Figure 32a. Such a decrease in current observed in Figure 32a is explained by the progressive growth of SiO_2_ at the junctions between NWs as indicated by the passage from step (1) to (1.a) in Figure 32b. With a time constant of 2.2 days, the silica shell formed is thick enough to electrically isolate the NW–NW junctions and prevent current flow through the nanonet. The use of the SiNNs is then jeopardized as a functional device under air. Nevertheless, when low-temperature annealing (400 °C) is performed just after removal of SiO_2_ by treatment with hydrofluoric acid (HF), the conductance shown in Figure 32a decreases by only 20% and then stabilizes for several months [48].

HRTEM analysis, presented in Figure 33a.1 with its equivalent diagram (a.2), reveals the presence of a neck at the junction between two NWs. On this micrograph, the visible dislocation shows a continuity of crystalline planes between the two NWs. This crystal lattice continuity explains the stabilization of the current observed in Figure 32a even after several months of exposure to air. The 20% decrease in current after a few days is explained by the progressive reduction of the neck by the growth of the native oxide. However, when the reoxidation stops for a SiO_2_ thickness of about 2 nm, the neck size stabilizes and the current is then constant even under air (Figure 32b, step (1.b)).

### 8.2. Surface and Interfaces

#### 8.2.1. Modulation Thanks to Interface Surrounding the Channel

The quality of the interface surrounding the channel is a key parameter for all electronic devices such as FETs as it influences many parameters, from high to low frequency noise [191,192]. For example, a mechanism for filling or emptying traps at the NW/oxide interface leads to a memory effect as this gate dependency could compromise the operation of the transistor. According to the literature, the most commonly used oxides for passivation of silicon nanowires are silicon oxide (SiO_2_), aluminium oxide (Al_2_O_3_), and hafnium oxide (HfO_2_) [47,193]. It has been shown that Al_2_O_3_ offers better chemical stability than silicon oxide while maintaining excellent sensitivity in the case of using them as sensors [194,195].

In their early work, Legallais et al. studied SiNN FETs passivated by the native silicon dioxide layer which systematically grows when the silicon is exposed to air [47]. It is known this native SiO_2_ provides poor-quality interfaces and induces the formation of a high density of dangling bonds at the interface. On the basis of the foregoing, they opted for an alumina encapsulation layer as it is fully compatible with the integration process and can be easily etched before contact deposition, using HF treatment [47]. Atomic layer deposition (ALD) was chosen for alumina deposition since it involves a self-limiting growth mechanism that enables the formation of high quality and homogenous thin film. Moreover, this technique provides a conformal coating and properly encapsulates SiNWs while preserving the sintered NW–NW junctions (Figure 34).

As native SiO_2_ has a high density of dangling bonds at the interface, this interruption in the periodic lattice structure acts as interface traps for carriers. These traps are responsible for the reduction of transistor performances. On the contrary, alumina provides a better quality interface and improved electrical characteristics, as clearly shown in Figure 35A,B.

As a result, this study shows that proper material engineering of nanonets via alumina encapsulation can drastically enhance the electrical characteristics of back-gate FETs.

At the same time, although silicon nanowires are considered promising for future biomedical sensors, their limited stability under physiological conditions is a challenge for sensor development. Solving this issue by surface engineering as described above opens up new possibilities for sensor improvements [81,196,197].

However, as with thin films, the nanostructures composing the nanonet can be functionalized with, for example, molecules or proteins. DNA sensors based on SiNN have been achieved by chemically modifying the surface of nanonet with 3-aminopropyl-triethoxysilane (APTES) silanization [28] or (3-glycidyloxypropyl)trimethoxysilane (GOPS) [198]. Particularly, GOPS functionalization enables DNA electrical detection with SiNN-FETs [199]. Additionally, silicon nanonet field effect transistors (SiNN-FETs) were biomodified using thrombin-binding aptamer (TBA-15) [200] with the aim to detect thrombin protein. As an illustration, Figure 36 shows the shift in threshold voltage induced by aptamer probes grafting on the SiNN surface. As a result, again it has been demonstrated that the surrounding material can drastically change the characteristics of the SiNN-FETs. 

#### 8.2.2. Modulation Thanks to Functionalization under the Channel

As previously stated, the complex electronic mechanisms involved in electronic devices are highly dependent not only on the energy barriers to conquer but also on the interface traps, which can induce modification of the charge carrier density thus changing the performances of the devices. The phenomena are poorly understood, and the roots of changes are numerous ranging from the neutralization of surface defects, the modification of the surface energy, and even the creation of interface dipoles. Celle et al. focused on the control of the chemical nature of the interface between the gate oxide and the semiconductor, the place where the conduction channel is established in FETs (back gate-bottom contact structure Figure 37) [201].

They show that thiolated self-assembled monolayer (SAMs) can be used to anchor source-drain gold electrodes on the substrate, leading to excellent electronic performances of the organic field-effect transistor (OFET) on the same level as those using a standard electrode process. It has been demonstrated that the threshold voltage is tunable while keeping the other electrical properties nearly unchanged by functionalization of the surface of the substrate under the channel (Figure 38). These self-assembled monolayers strongly modify the OFET characteristics, leading notably to charge carrier mobility [201,202].

SAMs are known to generate a built-in electric field, which modifies the carrier density in the transistor channel. For instance, it is seen that fluorinated SAMs have the tendency to generate a local electric field that accumulates holes, and on the contrary, devices with 3-aminopropyl-triethoxysilane (APTS), or triarylamine triethoxysilane derivative (TAATS) accumulate electrons, leading to the need for a very large negative gate bias to turn on organic semiconductor which is polytriarylamine (PTAA), into hole accumulation mode. They have shown that this local electric field is related to the dipole moment of the molecules [201,202].

### 8.3. Silicidation 

Silicidation was commonly used since the 1980s in MOSFETs in forming an alloy between metal and silicon during an annealing process in order to decrease the electrical contact resistance between the two materials. During this step, solid-state reactions occur by diffusion and/or nucleation processes of the thermally activated species. Depending on the experimental annealing conditions, different stable crystalline phases called silicides can form and are denoted M_x_Si_y_ with M the metal used. Silicon has the particularity of associating with many metals, some of which are grouped in Table 3. Historically, the silicides TiSi_2_, CoSi_2_, and more recently NiSi have been the most used by the microelectronics industry for their low electrical resistivity ranging from 10 to 25 μΩ.cm [203]. Nevertheless, Ti- and Co-based alloys require higher temperature annealing which may be responsible for the deactivation of the dopants contained in the silicon and implies an increase in contact resistance at the interface between silicide and Si [204]. Moreover, for some Co silicides, Si is the dominant diffusion species, which causes the formation of gaps or voids in the silicon. Following this phenomenon, the diffusion of Si gaps can also become considerable. Thus, simultaneous flows of matter in one direction and of vacancies in the other lead to the appearance of porosities within the silicon, a phenomenon commonly known as the Kirkendall effect. This is probably one of the reasons why a high increase in the electrical resistance of Co silicides is observed when the channel length of MOSFETs is below 50 nm [205,206].

For the reasons stated above, NiSi silicide is today widely used due to its low electrical resistivity (10.5–18 μΩ.cm) as well as its lower formation temperature (400 °C). Furthermore, Ni-based silicidation is a diffusion process in which Ni is the dominant diffusion species [206]. While, contrary to Co silicides, the gaps generated during thermal annealing are located in the metallic contact and avoid the appearance of porosities within the silicon. Nevertheless, the complexity of the Ni-Si phase diagram presented in Figure 39 indicates the difficulty to control, during annealing, the formation of the desired NiSi phase. Indeed, many crystalline and stable phases at room temperature are likely to form [209]. Note that the variations in resistivity from one phase to another are very large, ranging from 7 to 150 μΩ.cm (Table 4).

Currently, the salicidation process [210] (a portmanteau for ‘self-aligned silicide’) is used to achieve simultaneous silicidation of the source, drain. This technique consists in forming the Ni_2_Si phase by diffusion of Ni during the first annealing between 270 and 350 °C. Then, after etching the excess of unreacted nickel, the NiSi silicide is formed after second annealing between 400 and 550 °C [211]. According to Ottaviani, NiSi can only be formed after consumption of the entire metal reservoir, which explains the need to etch the excess Ni after the first annealing [212].

It was shown that the NiSi phase can form a low resistive Si-Ni interface for a thermal budget compatible with the specifications of today’s microelectronics industry.

#### Silicidation of Silicon Nanowires with Nickel

For silicon nanowires, the formation of a silicide also makes it possible to form an abrupt interface over the entire cross-section of the nanowire between the silicide and the nanowire, which facilitates the injection of the carriers. Indeed, the nickel penetrates over the entire volume of the SiNW and then diffuses longitudinally. Otherwise, for non-silicided nanowires, the injection of the carriers is less efficient since it is done radially through the edges of the SiNWs. In addition, Chou et al. studied the kinetics of NiSi formation between 450 and 750 °C by in-situ transmission electron microscopy analyses [213,214].

Nevertheless, the NiSi phase is often accompanied by a Ni-rich phase that forms at lower temperatures [215,216,217]. Other crystallographic studies have shown the possibility of obtaining Ni-rich phases such as Ni_31_Si_12_ and Ni_2_Si for temperatures between 400 and 450 °C [176,218]. More surprisingly, NiSi_2_ silicide has been stabilized at low temperatures between 440 and 550 °C while it forms at about 800 °C for bulk materials [175,176,218,219,220]. In order to evaluate the phases likely to form for a given temperature, Figure 40 represents the minimum and maximum temperatures of formation of the different Ni_x_Si_y_ silicides in silicon nanowires from references reported in the literature.

According to Figure 40 based on SiNWs, for an annealing temperature of about 400–450 °C, we can see that the Ni_31_Si_12_, Ni_2_Si, NiSi, and NiSi_2_ phases are likely to form, i.e., a variation in electrical resistivity between 10.5 and 150 μΩ.cm (Table 4). Thus, obtaining the NiSi phase in SiNWs is complex and the silicidation involves many parameters such as the crystal orientation [176,218], the diameter of the nanowire [217], the thickness of the nickel film [225], or the thickness of the oxide surrounding the nanowire [221]. According to the work of Ogata et al. [220], the diffusion length of silicide in SiNW evolves with the square root of the annealing time.

According to the literature carried out previously, the desired NiSi silicide of low electrical resistivity is formed at a temperature of about 400 °C for massive materials (Table 4). In the context of NW silicidation, the complex growth mechanisms involved do not allow to conclude on an optimal temperature and time for the formation of the NiSi phase despite a rich literature on this subject. Nevertheless, according to the work of Ternon et al. [48] and the studies of Byon et al. [182], an optimum of the electrical properties can be distinguished for annealing at 400 °C which is consistent with the NiSi formation temperature. Moreover, this temperature does not exceed the maximum temperature allowed for integration on the back-end of a reading circuit.

To complete the silicidation study, the impact of silicide existence on the electrical characteristics of SiNWs was also considered [226]. The main advantage of silicide contacts over non-silicided contacts is better device performance by modifying the nature of the metal–semiconductor interface, thus reducing series resistance [11,214,215,227]. Guillaume Rosaz showed that the NiSi silicide, formed at 400 °C in the nanowires, lowers the height of the Schottky barrier, thus favoring the injection of the carriers. The performance of the transistors is significantly improved by an increase in current accompanied by a decrease in subthreshold swing [3,11]. The thermally activated intrusion of nickel into the SiNW is an inexpensive and robust, but not fully controlled, process. They demonstrated that the silicidation reaction seems to be self-limited (see Figure 41c).

The silicide region lessens the resistance of the contacts in two ways. First, the low sheet resistance of the silicide layer reduces the in-plane contact resistance, and second, the silicide reaction leads to an intimate and more reliable metal–semiconductor (MS) contact and transforms the contact surface initially on the surface of the NW into a contact surface on the cross-section of the NW. As a result, in terms of performance, the most substantial properties of a silicide material are its low electrical resistivity and its lower Schottky barrier height [227,228]. Byon et al. also studied the impact of silicidation on the electrical properties of transistors and observed an optimum between 400 and 450 °C after successive annealing from 250 to 500 °C [182].

Figure 42 presents the transfer characteristics before and after annealing of a 20 μm channel SiNN-FET elaborated with a density of 42 × 10^6^ NWs cm^−2^ (drain-source bias of −4 V). Before annealing, the device exhibits and the state of 9 nA and 0.7 pA respectively, resulting in an on/off ratio of 10^4^. After annealing, the state reaches 170 nA while the state remains constant, corresponding to an enlarged on/off ratio >10^5^ Such improvement by one order of magnitude confirms the formation of a low resistive phase, enhancing hole injection at the metal/SiNW contact.

## 9. Applications

Silicon nanowires can be exploited in many ways in electronic devices and can find numerous real applications such as displays, data storage, 3-D computing, lasers, smart cards, wearable electronics, high efficiency programming, ring oscillators [6,31,168]. They have so far shown promising applications in areas ranging from biological sensors, thermoelectric convertors, opto-mechanical devices, piezoelectric sensors, and solar cells among others. Describing all the potential applications is detail is far beyond the scope of this review. However, dealing with devices without describing applications would leave a taste of unfinished business. As a consequence, in this section, without being exhaustive, we choose to summarize some basic applications for the simplest SiNW-based devices (resistor, transistors, and diodes).

### 9.1. Photodetectors

Optical properties of SiNWs have allowed envisaging the use of SiNWs as optically active materials for photonics application. The ease of bandgap conversion from indirect to direct band due to dimension, crystallography, mechanical strain, and alloying allows SiNWs to be used in the optical applications—e.g., photodetectors (PDs) and light emitters (LEs). Since silicon nanowires have a superior ability to tune absorption with morphology, Um et al. reported that SiNWs with a coating of an indium oxide layer on it, lead to efficient carrier separation and collection, resulting in an improvement of quantum efficiency and by controlling the nanowire radii, can create a multispectral detector [229].

### 9.2. Memories

A self-alignment technique can be used to position the silicon nanowires, which could allow for lower production costs than current flash memory cards. SiNW-based memory devices showed better stability at higher temperatures, no power consumption in the off-state, and very small switching energy (10^−14^ J) [230].

With the advent of new CMOS compatible fabrication methods for silicon nanowires, it is now feasible to build memory and memristive devices. The SiNW-FET based dielectric charge-trapping flash-like memories have been fabricated and fully characterized by Zhu et al. (Figure 43c,d) [231].

These non-volatile memory devices exhibited fast programming/erasing speed, excellent retention, and endurance, indicating the advantages of integrating the multilayer of charge-storage stacks on the nanowire channel. Such high-performance flash-like non-volatile memory can be integrated into the microprocessor chip as the local memory which requires high density and good endurance [231,232,233].

### 9.3. Biosensors

Sensors are important tools for life sciences and biochemistry. The use of sensors in these areas leads to the detection and diagnosis of diseases and to the discovery of new drugs [17,71]. Microelectronic sensors based on thin film transistors and ion-sensitive filed-effect transistors (ISFETs) are used since the 1970s. They offered a cheap solution over the chemical sensors and could be integrated on a chip [71]. Nevertheless, the microelectronic sensors did not have the required characteristics to be used as biosensors because of their undesired sensitivity to temperature and light and because their parameters were not fixed over time. Additionally, solid-state electrodes were not reliable and this led to the use of chemical sensors [71]. Currently, the detection of biomolecules at low concentrations is achieved with fluorescent labeling and optical detection methods. However, this technique is expensive and time-consuming and thus silicon nanowires may be an alternative solution [71,235].

Silicon nanowires can almost act as perfect biosensors due to their inherent properties [235]. When used as sensors, their characteristics include ultrahigh sensitivity because the molecule being sensed depletes or accumulates the charges in the bulk of the nanowire [17,235]. In addition, direct label-free detection allows the molecules to be detected in real time which eliminates the time consuming labeling chemistry [17,23,71]. Another important characteristic is that they are non-radioactive and that sensor arrays can allow detection of different molecules in the same solution [23,71].

Although SiNW sensors have great characteristics there are factors that can affect their performance. These include the surrounding environment of the nanowire and the electrostatic screening action of the ions in the solution. The performance of a sensor can be characterized by sensitivity, settling time, and selectivity. A simulation with a silicon nanowire with a length between 2 and 20 μm showed how the sensitivity can be influenced by several factors. The results showed that the sensitivity increases with reduced dopant density and decreased diameter and length. However, it is not possible to reduce these quantities as much as needed to achieve the maximum desired sensitivity due to the dopant fluctuations effect. Furthermore, this simulation revealed that the dielectric constant of the surrounding environment affects the sensitivity of SiNW. If the surrounding media is air then the sensor has greater sensitivity if it is designed to operate in the depletion mode. Finally, parasitic ions in the surrounding solution of the sensor screen the charge of the target molecule and reduce sensitivity [71].

Experimental applications of SiNW sensors include the detection of proteins, DNA, pH, drug discovery, single viruses, glucose, and arrays for parallel molecule detection, as shown in Figure 44 [17,61,236].

### 9.4. Gas Sensor

Silicon nanowires can find applications as gas sensors too. As an example, this was demonstrated for H_2_ sensing when the surface of an n-type SiNW was coated with palladium nanoparticles (Pd particles). Good selectivity was demonstrated as no response was observed when exposed to NH_3_ or N_2_O gases. However, when H_2_ gas flowed over the sensor, the current flowing through was increased, as shown in Figure 45. Furthermore, the SiNW sensor responded faster (2.3 s) than an ordinary macroscopic Pd wire sensor (more than 10 s) [236]. 

### 9.5. Thermoelectric Application

Thermoelectric devices can convert heat directly into electrical power, and vice versa, and they have a broad range of applications: energy recovery and green energy harvesting; energy micro-harvesting (scavenging) for the capillary supply of small systems, such as sensor nodes for Internet of Things (IoT); powering of systems in remote and harsh environments typical, for example, of spatial exploration; localized and optimized cooling of small systems, where the reliability and the compactness can play a fundamental role [101,237,238]. Unfortunately, the available materials with thermoelectric properties good enough for an acceptable thermal-to-electrical conversion efficiency limit all of the potentialities offered by thermoelectric devices at the current state of the art.

The observed high electrical and low thermal conductivity of highly doped SiNWs arrays, shows that SiNWs arrays represent a promising material for thermoelectric applications. Hence, it could make a significant contribution in the fundamental fields of energy micro-harvesting (scavenging) and macro-harvesting [101,237,238].

## 10. Conclusions

Semiconductor nanowires, especially silicon nanowires, have aroused a lot of scientific interest over the past 25 years and have been considered a promising material for nanoscale devices and integrated circuits. Key parameters in realizing application through a bottom-up paradigm include chemical composition, structure, size, morphology, and doping which have been fully controlled in semiconductor NW systems. Among semiconductor NWs, silicon NWs—owing to their unique physical and chemical properties—show promise for a wide range of applications, including FETs. However, such devices based on single SiNWs face complicated fabrication processes and low reproducibility of their electrical characteristics which originate from various issues such as dispersion in length, diameter, and doping of individual nanowires, wire surface passivation, poorly controlled gate length, or silicidation with the silicide length and quality arising from annealing step.

Researchers tried to solve these issues with multi-channel SiNW-based devices which have better stability and reproducibility in comparison with single SiNW-based devices. Apart from increased on-current in MPC-FETs, some other parameters include on-to-off ratio or subthreshold swing degrade. This degradation occurs due to intrinsic disparities of all parameters which have already been mentioned for single SiNWs such as length, diameter, doping, and surface properties.

Silicon nanonets are networks of randomly oriented silicon nanowires. Due to its flexibility, transparency, and reproducibility, this material is highly attractive as an alternative to amorphous silicon or organic materials for various macro-electronic applications involving sensors and displays. Based on the easy integration process simply relying on standard photolithography, it has demonstrated successful workability, reproducibility, and excellent air stability along with interesting performance for device channel length ranging from micrometer to millimeter scales. SiNN-FETs also displayed good stability and reproducibility which differentiates them from single SiNW- and MPC-FETs and which are the results of averaging effect over thousands of nanowires and their junctions.

According to literature, the nanonet-based thin film transistor technology has the respective advantages of poly-Si, a-Si, or organic material, without their particular drawbacks, along with being flexible. Moreover, low-cost and large-scale technology is an important asset of this material, in comparison to the issues surrounding the poly-Si option. Furthermore, using nanostructures—such as a network of nanowires—allows electronics to be smaller, more powerful, and more efficient. Finally, in conclusion, nanonets can address the need for cost-effective, reproducible, and efficient systems to exploit nanoscale properties while being easily manipulated and compatible with large-scale integration and opening up many opportunities, both in terms of applications and fundamental studies, in short, a new field of research.

## Figures and Tables

**Figure 1 nanomaterials-12-01043-f001:**
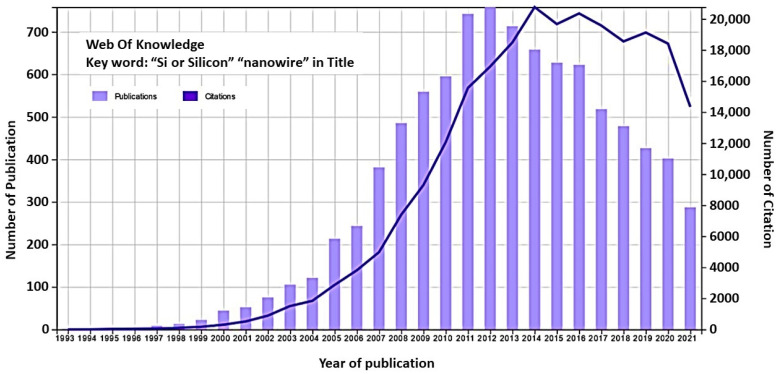
Number of articles published per year and the number of citations based on a search using the keyword “Si or Silicon Nanowire” on the website Of Knowledge.

**Figure 2 nanomaterials-12-01043-f002:**
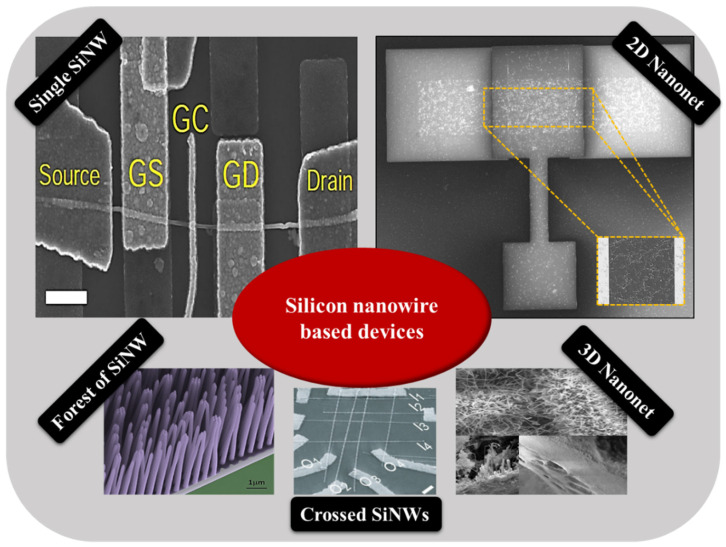
Different types of nanowire-based samples [33,34].

**Figure 3 nanomaterials-12-01043-f003:**
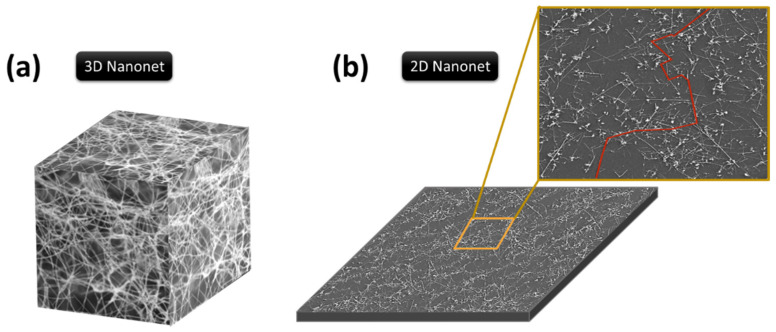
Schematic made by SEM images for representation of a nanonet (**a**) three-dimensional (3D nanonet) and (**b**) two-dimensional (2D nanonet). A top view and percolation path in red are highlighted in image (**b**).

**Figure 4 nanomaterials-12-01043-f004:**
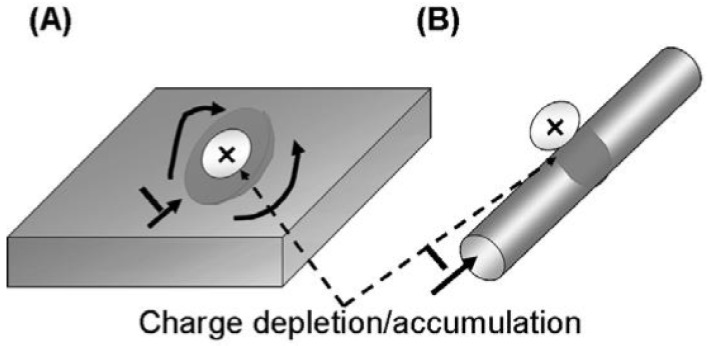
The major advantage of 1D nanostructures (**B**) over 2D thin film (**A**). Binding to 1D nanowire leads to depletion or accumulation in the ‘bulk’ of the nanowire as opposed to only the surface in the 2D thin-film case. Reproduced from [61].

**Figure 5 nanomaterials-12-01043-f005:**
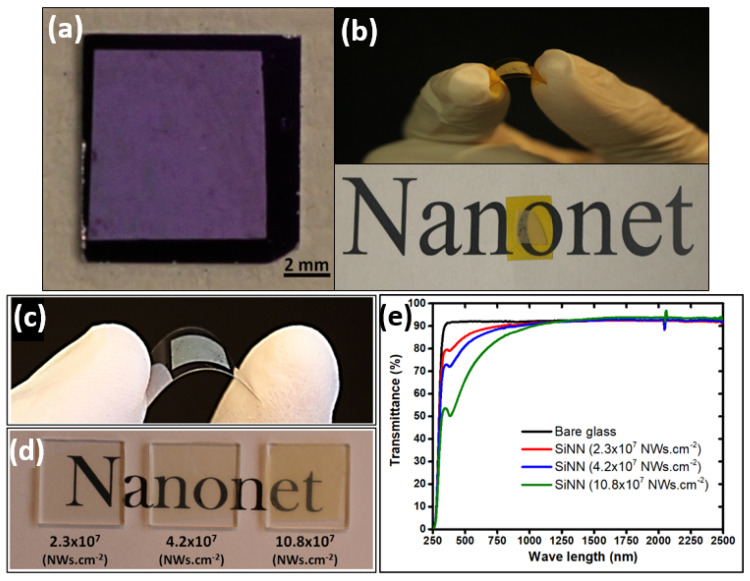
Silicon nanowire nanonets manufactured by vacuum filtration and then transferred to (**a**) Silicon/Si_3_N_4_, (**b**) Kapton, (**c**) plastic, and (**d**) glass substrates. (**e**) The transmittance of SiNN with the three densities were shown in image (**e**), the transmittance of the substrate (bare glass) is also reported. Reproduced from [114].

**Figure 6 nanomaterials-12-01043-f006:**
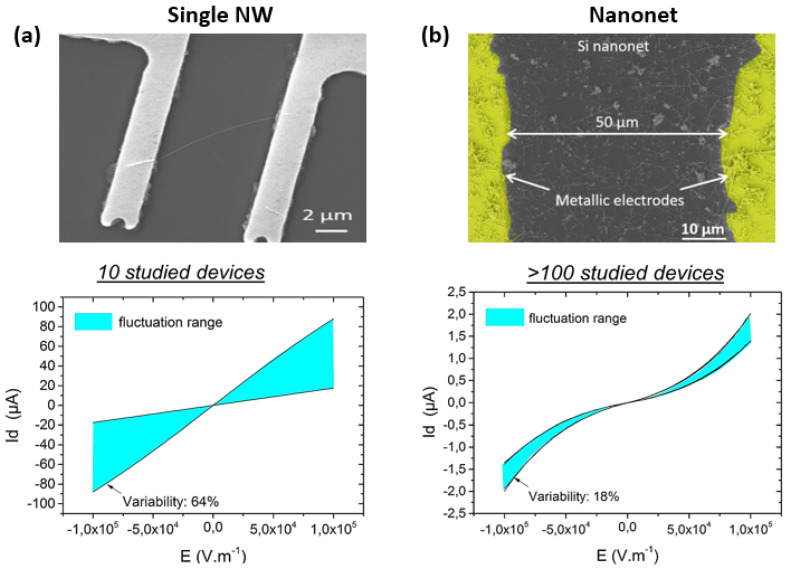
SEM image and fluctuation range in the current of (**a**) single nanowire and (**b**) nanonet-based resistors. Reproduced from ref. [28].

**Figure 7 nanomaterials-12-01043-f007:**
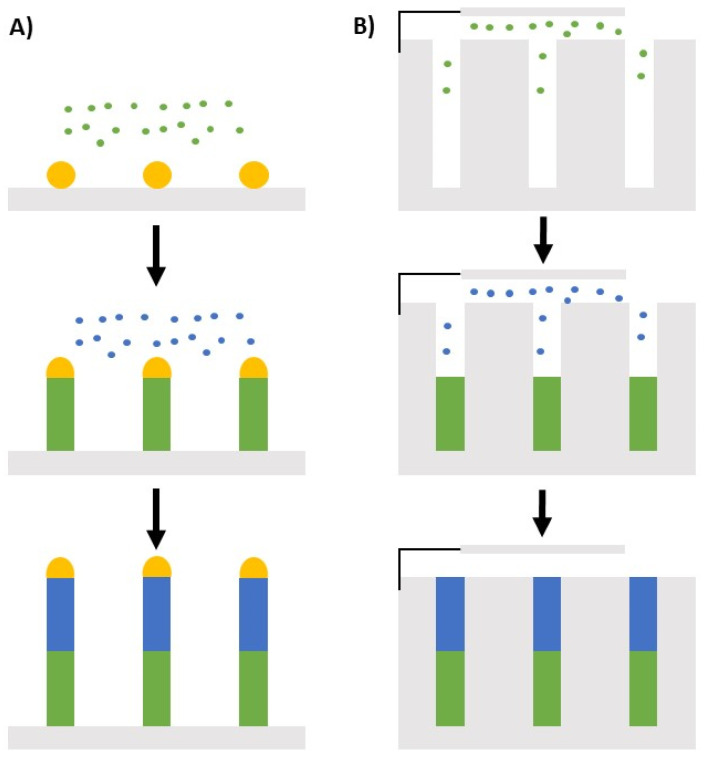
Schematic representation of two bottom-up synthesis methods. (**A**) Vapour-Liquid-Solid Chemical Vapor Deposition (VLS-CVD). Segmentation in composition is possible by the modulation of the gaseous precursor. (**B**) Electrochemical deposition in solution into anodic aluminum oxide. Segmentation is also possible.

**Figure 8 nanomaterials-12-01043-f008:**
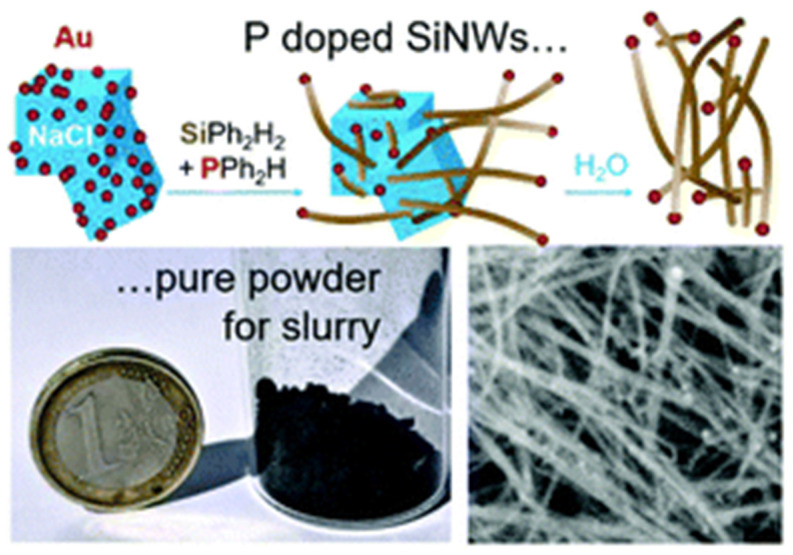
Schematic of the growth principle of SiNWs, and SEM image of as grown silicon nanowires Reproduced from [125].

**Figure 9 nanomaterials-12-01043-f009:**
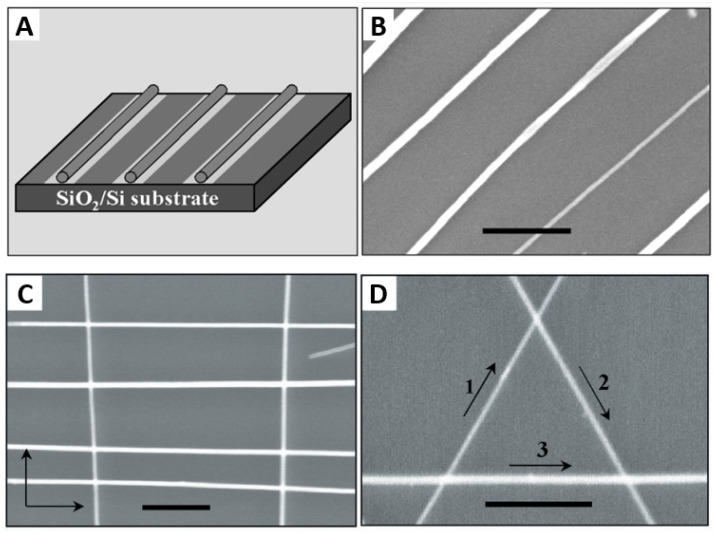
Assembly of periodic NW arrays and layer-by-layer assembly of crossed NW arrays. (**A**) Schematic view of the assembly of NWs onto a chemically patterned substrate. The light gray areas correspond to NH_2_-terminated surfaces, whereas the dark gray areas correspond to either methyl-terminated or bare surfaces. NWs are preferentially attracted to the NH_2_-terminated regions of the surface. (**B**) Parallel arrays of GaP NWs with 500-nm separation obtained with a patterned SAM surface. (**C**) Typical SEM images of crossed arrays of InP NWs obtained in a two-step assembly process with orthogonal flow directions for the sequential steps. Flow directions are highlighted by arrows in the images. (**D**) An equilateral triangle of GaP NWs obtained in a three-step assembly process, with 60° angles between flow directions, which are indicated by numbered arrows. The scale bars correspond to 500 nm in (**B**–**D**) Reproduced from [130].

**Figure 10 nanomaterials-12-01043-f010:**
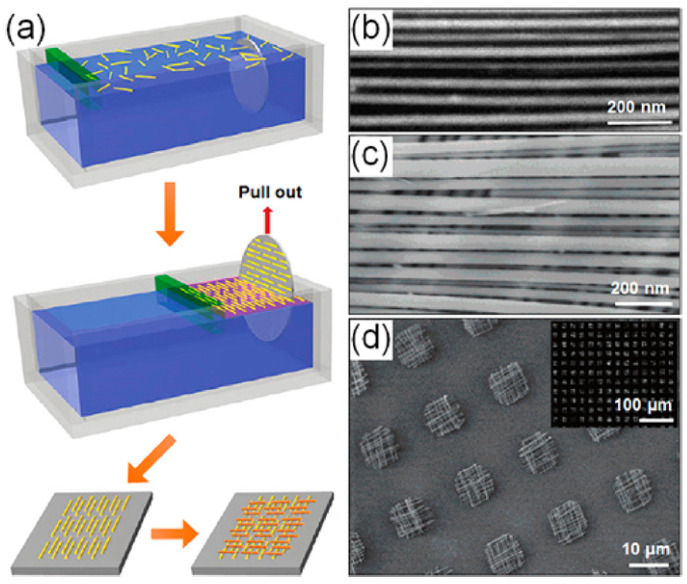
Langmuir−Blodgett assembly of nanowires. (**a**) Schematic illustration of the Langmuir−Blodgett assembly process. Reproduced from [134]. (**b**,**c**) SEM images of a high-density parallel nanowire array (**b**) and crossed nanowire array (**c**) on the substrates. (**d**) SEM images at different magnifications for patterned crossed nanowire arrays. Reproduced from [43].

**Figure 11 nanomaterials-12-01043-f011:**
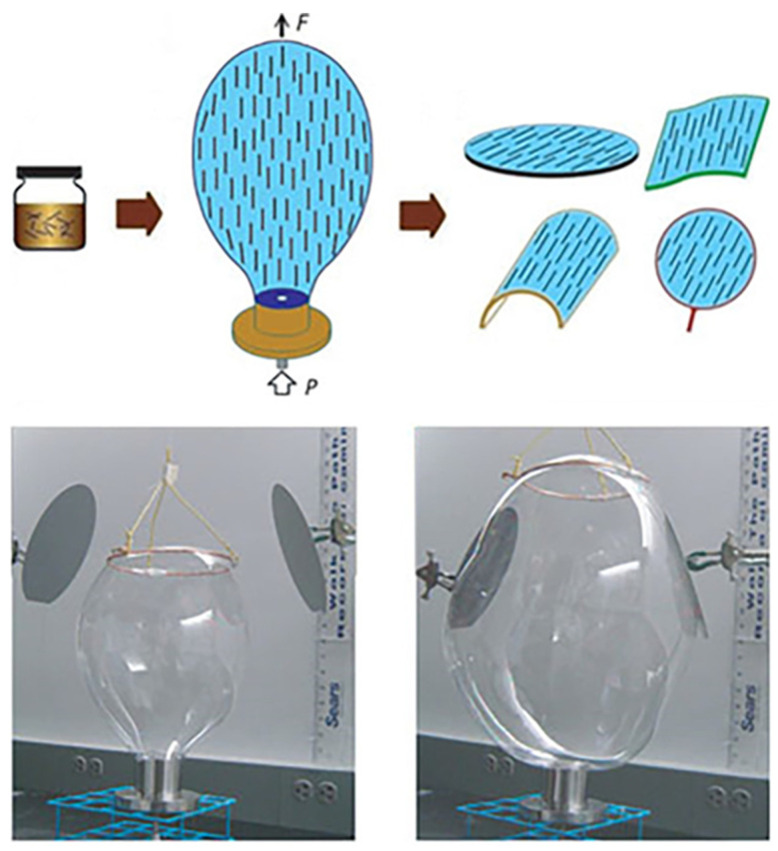
Illustration of blown bubble films method and photographs of the directed bubble expansion process in the early and final stages. Reproduced from [137].

**Figure 12 nanomaterials-12-01043-f012:**
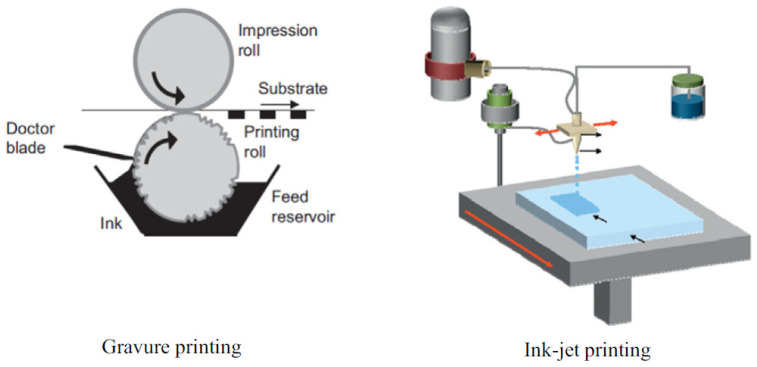
Illustration of printing apparatus. Reproduced from [139].

**Figure 13 nanomaterials-12-01043-f013:**
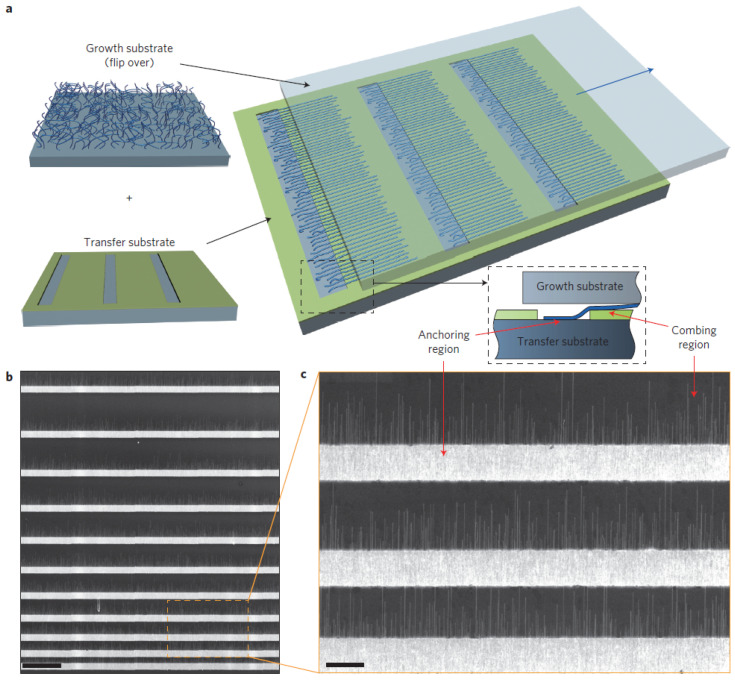
Schematics and demonstration of nano-combing. (**a**) Schematics of the nano-combing process. The blue arrow indicates the traveling direction of the growth substrate with respect to the target substrate, which yields a combing/aligning force that is parallel and opposite to the anchoring force. The dashed window at the right bottom shows a side view of the nano-combing process. (**b**,**c**) SEM images of silicon nanowires on the combing (resist) surface at different magnifications. The thickness of the resist (S1805) layer was 70 nm. Scale bars: 50 μm (**b**), 10 μm (**c**) Reproduced from [144].

**Figure 14 nanomaterials-12-01043-f014:**
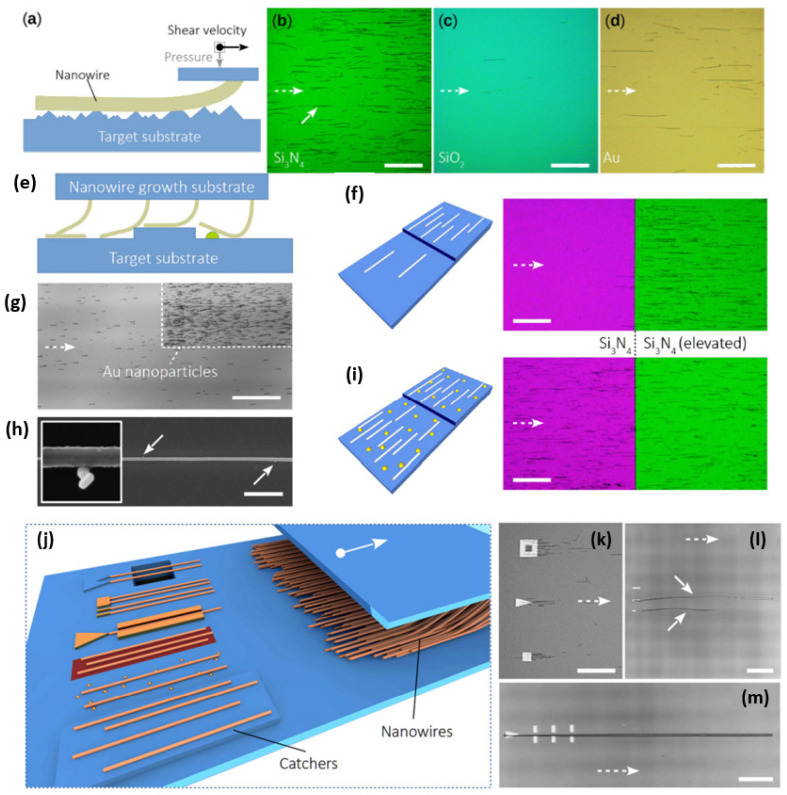
(**a**–**d**) Friction-based SCCP concept. (**a**) Schematic illustration of a nanowire in mechanical contact with a surface. The frictional force is predominantly influenced by the shear velocity vector, applied load, contact morphology, and materials of the nanowire and the target substrate. (**b**–**d**) Optical microscopy images of nanowires transferred in a lubricant-free manner (exemplified by the solid arrow in (**a**)) onto Si_3_N_4_, SiO_2_, and Au surfaces. The dashed arrow represents the shear direction of the growth substrate. (**e**–**i**) Influence of local surface features on SCCP. (**e**) Schematic illustration showing the interaction of a nanowire with a previously deposited nanowire (left), the interactions of nanowires with a step (center), and the interaction of a nanowire with a nanoparticle (right). (**f**) Schematic and optical images of an 80 nm step in Si_3_N_4_, revealing that nanowires are preferentially deposited onto the elevated area. (**g**) Optical micrograph depicting an area decorated locally with Au nanoparticles of 50 nm in diameter. The nanoparticles increase the frictional force acting on the nanowires and, therefore, significantly increase the deposited nanowire density. (**h**) SEM image of a nanowire on a surface decorated with Au nanoparticles of 20 nm in diameter. The arrows indicate the positions of nanoparticles. The inset shows a magnified region containing a nanowire and Au nanoparticles. (**i**) The effect of the step, as shown in (**f**), is masked when Au nanoparticles (here, 50 nm in diameter) are present. The shear direction of the growth substrate is indicated in all images by a dashed arrow. The scale bars for (**f**–**i**) represent 100 μm, and that for (**h**) represents 1 μm. (**j**–**m**) Towards SCCP nano-device fabrication. (**j**) Schematic illustration of various catcher concepts, listed from front to back: elevated plateaux, nanoparticles, changes in surface roughness or material composition, catchers with selectivity or guiding rails, catchers for single and multiple-nanowire positioning, and catchers fabricated out of the substrate material with nanowires spanning a trench. (**k**) Catchers on SiO_2_ of different lateral shapes, leading to an increased nanowire density adjacent to the catcher. The dashed arrow indicates the shear direction of the nanowire growth substrate. (**l**) Gold catchers on Si_3_N_4_ with the ability to position single nanowires (see white arrows). The width of a single structure is 300 nm. (**m**) Optical image of one triangular and six rectangular Au structures for single-nanowire positioning. When the triangular structure catches a single nanowire, the rectangular structures appear to assist, serving a function similar to that of a guiding rail, as revealed by experiments. Reproduced from [143].

**Figure 15 nanomaterials-12-01043-f015:**
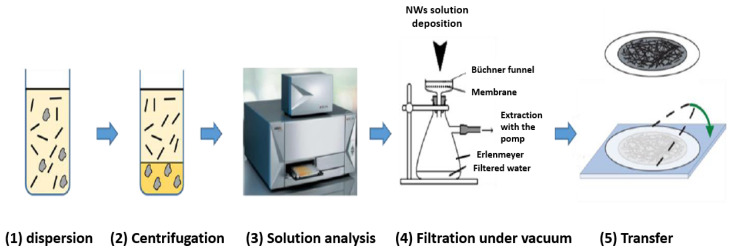
Overview of the protocol for manufacturing nanonets. This process consists of five main steps: (**1**) dispersion of the silicon NWs in solution, (**2**) purification of the NW suspension by centrifugation, (**3**) analysis of the suspension by absorption spectroscopy, (**4**) assembly of the NWs into nanonets by vacuum filtration, and (**5**) transfer of the nanonet onto a substrate. Reproduced from [27].

**Figure 16 nanomaterials-12-01043-f016:**
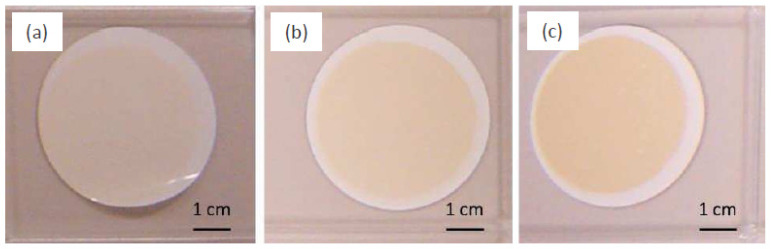
Membranes obtained after vacuum filtration of nanowires having the same arbitrarily fixed absorbance (0.06 at 400 nm) and different filtered volumes (**a**) 10 mL, (**b**) 20 mL, and (**c**) 35 mL. Reproduced from [28].

**Figure 17 nanomaterials-12-01043-f017:**
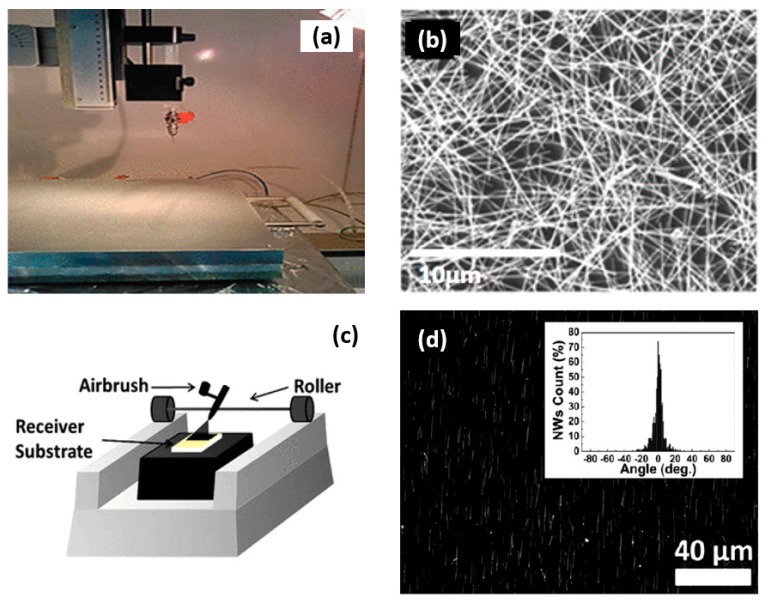
(**a**) Image of the electrostatic spray system step. Reproduced from [159]. (**b**) SEM image of the electrostatic spray deposited nanowire network. Reproduced from [159]. (**c**) Schematic of the spray coating apparatus. Reproduced from [158]. (**d**) representative dark-field optical image of spray-coated SiNWs on the SiO_x_/Si substrate and the constituent analysis of 700 SiNWs with respect to the flow direction. Reproduced from [158]. Spray coating method can be controlled under conditions of temperature, droplet size, spray coating angle, and airflow which makes this method interesting either in the well-aligned or well controlled density in large size of nanonet with a small lack in control over the low density of nanonets.

**Figure 18 nanomaterials-12-01043-f018:**
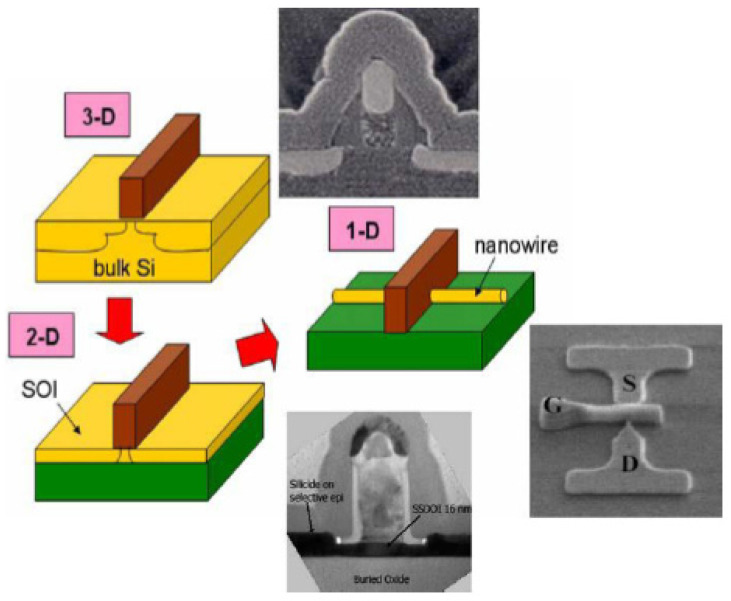
Continued scaling of silicon complementary metal–oxide–semiconductor (CMOS) transistor into nanometer regime requires the corresponding reduction in device active layer dimensionality. Reproduced from [6].

**Figure 19 nanomaterials-12-01043-f019:**
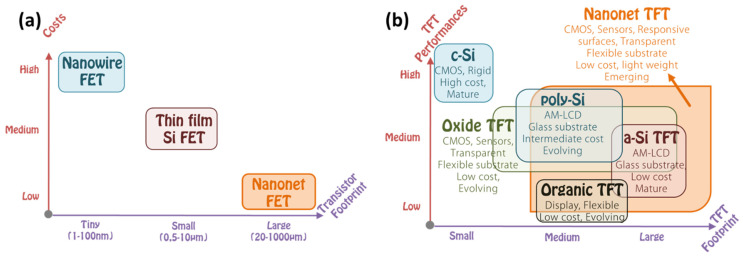
(**a**) Simple comparison between single SiNW-, thin film Si- and SiNN-FETs with respect to the industrial size and fabrication costs related. (**b**) Positioning (Nanonet TFT) in terms of thin-film transistor (TFT) performance versus footprint in comparison with the existing technologies (TFT based on: cSi: monocrystalline Si; poly-Si: polycrystalline Si; a-Si: amorphous Si; Organic: organic material; Oxide: metal oxide film).

**Figure 20 nanomaterials-12-01043-f020:**
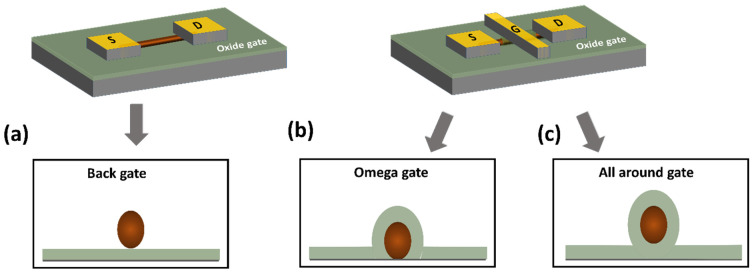
Schematic of NWFETS with (**a**) back gate, (**b**) semi-cylindrical top gate, and (**c**) cylindrical gate-all-around configurations. The nanowire is brown, gate-dielectric is light green, and source (S), drain (D), and top-gate (G) electrodes are gold. Insets show device cross section at the midpoint between source and drain.

**Figure 21 nanomaterials-12-01043-f021:**
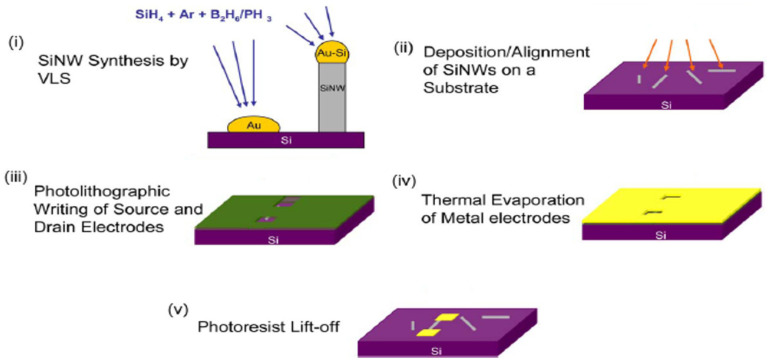
(**i**) The growth of SiNWs in CVD reaction via the VLS mechanism. **(ii**) Deposition/alignment of SiNWs on a silicon substrate. (**iii**) A photomask pattern to define source/drain electrodes. (**iv**) Thermal evaporation to deposit the source/drain contacts. (**v**) Lift-off the remaining photoresist with Remover PG. Adapted from [174].

**Figure 22 nanomaterials-12-01043-f022:**
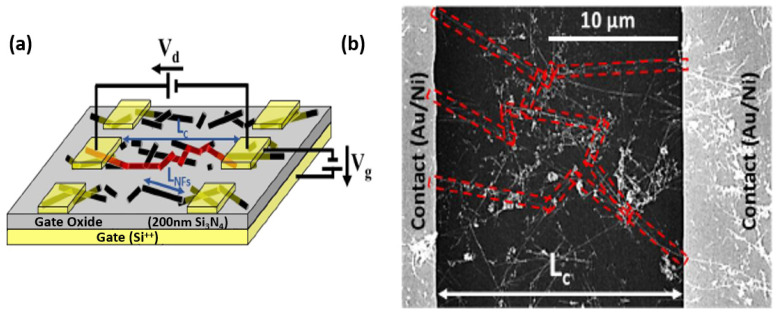
(**a**) Schematic and (**b**) SEM image of SiNN transistor studied. The schematic (**a**) refers to long channel transistors for which the length of the NWs (*L_NFs_*) is less than the channel length (*L**_c_*). Conversely, the SEM image (**b**) refers to the possible conduction paths involving NW–NW junctions are indicated in red. Reproduced from [27].

**Figure 23 nanomaterials-12-01043-f023:**
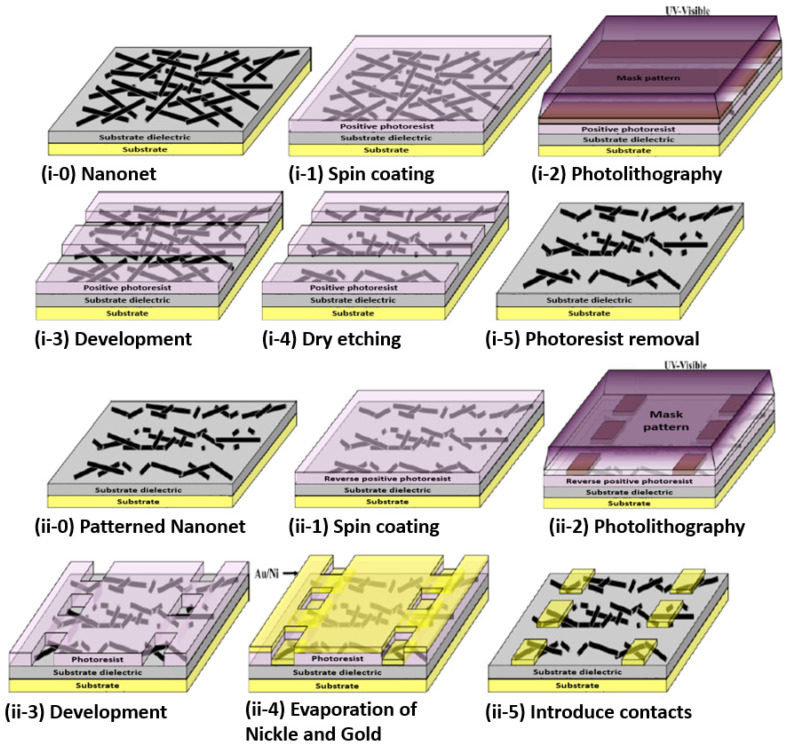
Main steps in the isolation of nanonets. (**i-0**) Fabrication of nanonets. (**i-1**) Deposition of the positive photoresist by the spin-coating technique. (**i-2**) Photolithography with the UV–visible through the mask. (**i-3**) Photoresist development. (**i-4**) Dry etching of the NWs by a sulfur hexafluoride plasma. (**i-5**) Photoresist removal. Main steps for the formation of source/drain contacts. (**ii-0**) Nanonet after isolation. (**ii-1**) Deposition of inversion photoresist using the spin-coating technique. (**ii-2**) Photolithography in the UV–visible through the mask aligned with the isolated nanonet. (**ii-3**) Photoresist development. (**ii-4**) Electron beam evaporation of nickel and gold. (**ii-5**) Revelation of the source/drain contacts after lifting of the photoresist (lift-off).

**Figure 24 nanomaterials-12-01043-f024:**
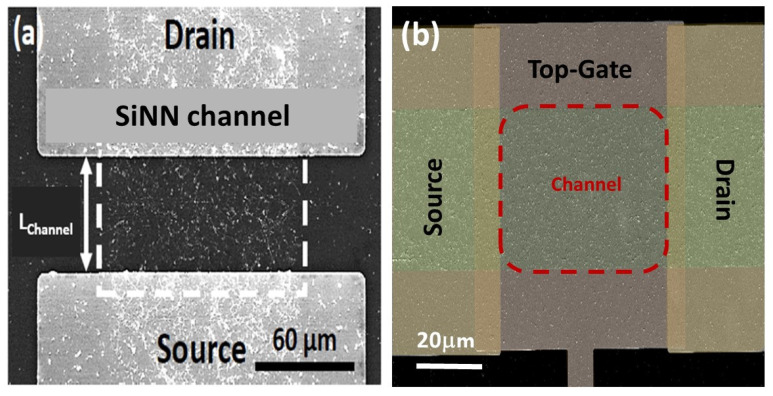
(**a**) SEM image final full back-gate FET based on SiNN. The length of the Si nanonet channel is 50 μm for a width of 120 μm. The square contacts measure 200 μm on each side. (**b**) SEM image final local top-gate FET based on SiNN. The length of the Si nanonet channel is 50 μm for a width of 100 μm. The square contacts measure 200 μm on each side. Although this integration process involves only simple and mastered steps, such integration has proven to be challenging since, to date, very few papers have presented the fabrication of this type of device. Moreover, to date, there are works based on the SiNN devices on rigid substrates, especially resistors and transistors but there is a great potential in the field of flexibility that is still unproven.

**Figure 25 nanomaterials-12-01043-f025:**
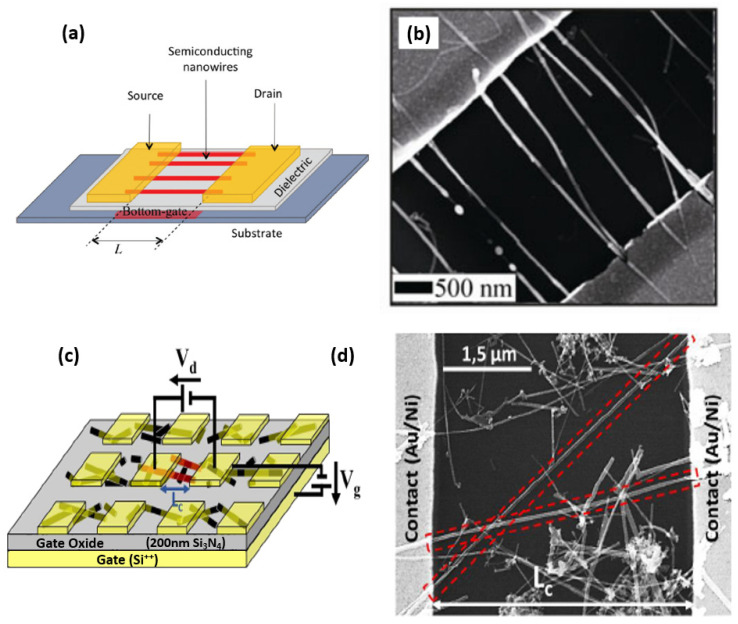
(**a**) Schematics of bottom-gate of multi parallel array silicon nanowires. Reproduced from [177] (**b**) SEM image of as fabricated parallel channels FETs with intruded NiSi_2_ Schottky barrier contacts. Reproduced from [178]. (**c**) Refer to the schematic of the short channel transistor (Nanowire length higher than channel length) (**d**) SEM image of MPC-FETs based on short-channel SiNN and the possible conduction paths involving with without-NW–NW junctions are indicated in red. Reproduced from [27].

**Figure 26 nanomaterials-12-01043-f026:**
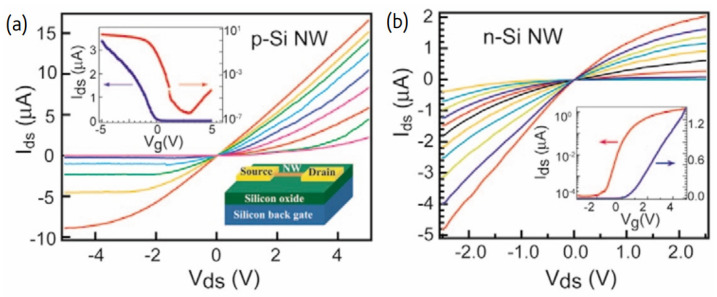
SiNW-FETs: a family of current versus drain-source voltage (I_ds_-V_ds_) plots for a representative (**a**) 20 nm p-Si NW device (channel length of 1 µm; from red to pink, V_g_ = −5 V to 3 V); and (**b**) 20 nm n-Si NW device (channel length of 2 µm; from yellow to red, V_g_ = −5 V to 5 V) in a standard back-gated NW-FET geometry as illustrated. Insets in (**a**,**b**) are current versus gate–voltage (I_ds_-V_g_) curves recorded for NWFETs plotted on linear (blue) and log (red) scales at V_ds_ = −1 V and 1 V, respectively. Reproduced from [31].

**Figure 27 nanomaterials-12-01043-f027:**
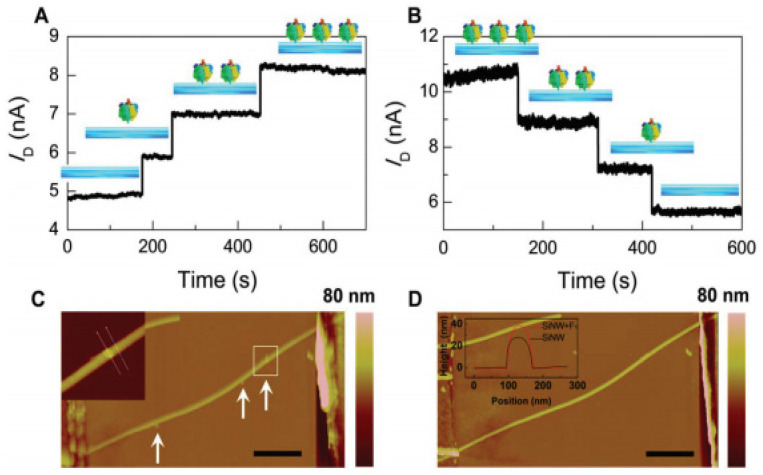
Sensing properties. (**A**,**B**) Real-time recordings of the absorption/desorption processes of F1-ATPases, showing the gradual changes in ID with three steps. Drain voltage, V_D_ = 0.1 V and gate voltage, V_G_ = 0 V. (**C**,**D**) Corresponding AFM images after protein delivery ((**C**), inset shows an enlarged image of a single F1 protein) and after further EDTA treatment ((**D**) inset is the height profile of the bare silicon nanowire and the nanowire with an adsorbed F1 protein particle in (**C**) inset. The total height of F1 is ~12 nm including a ~2 nm linkage). The scale bar is 1 µm. Reproduced from [75].

**Figure 28 nanomaterials-12-01043-f028:**
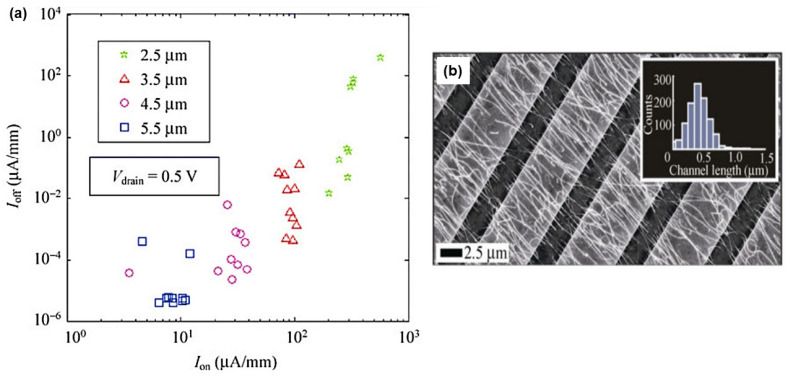
(**a**) Source-drain voltage V_sd_ versus a gate voltage V_g_ statistics of 36 nanowire parallel array FET devices. Each device consists of 500–1000 nanowires. Off-current versus on-current per mm electrode width for V_sd_ = 0.5 V. The on/off ratio is shown for devices with four different inter-electrode spacing but the same silicidation process (green/stars 2.5 µm; red/triangles 3.5 µm; purple/circles 4.5 µm; blue/sq. 5.5 µm). Reproduced from [178]. (**b**) High density nanowires are contacted by nickel electrodes. The inset displays the histogram of channel lengths of individual nanowires after silicidation for a device with 2.5 µm inter-electrode spacing. Reproduced from [178].

**Figure 29 nanomaterials-12-01043-f029:**
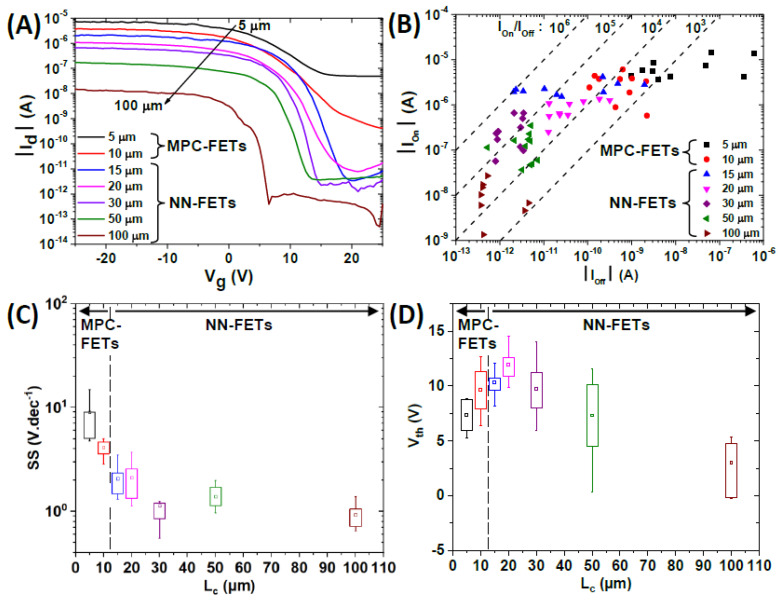
Study of transistor electrical properties for various channel lengths. (**A**) Typical transfer characteristics from 5 to 100 μm at V_d_ = −4 V. (**B**) On current (I_on_) as a function of off current (I_off_). The on current is defined as I_d_ at V_g_ = −25 V, V_d_ = −4 V, (**C**) subthreshold slope (SS), and (**D**) threshold voltage (V_th_) for various channel lengths extracted for about seventy transistors. For (**B**) the on-to-off ratio (I_on_/I_off_) is indicated by the dashed lines. Reproduced from [72].

**Figure 30 nanomaterials-12-01043-f030:**
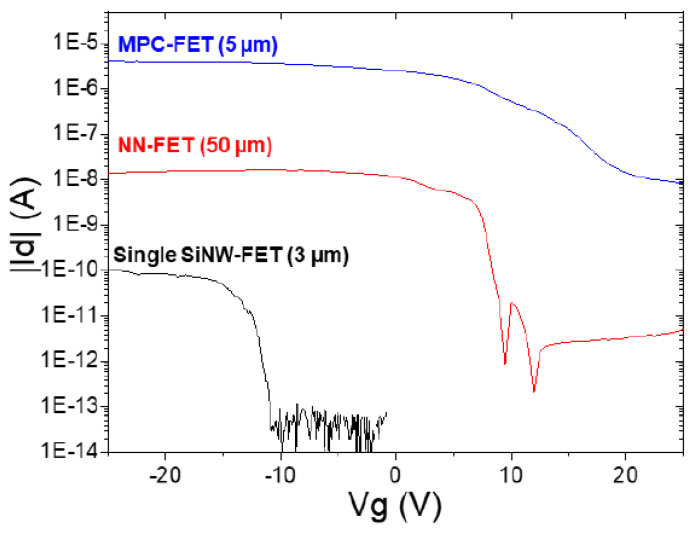
Typical transfer characteristic at V_d_ = −1 V for single SiNW-FET with *L_c_* = 3 μm (black), MPC-FET with *L_c_* = 5 μm (blue), SiNN-FET with *L_c_* = 50 μm (red). Reproduced from [72].

**Figure 31 nanomaterials-12-01043-f031:**
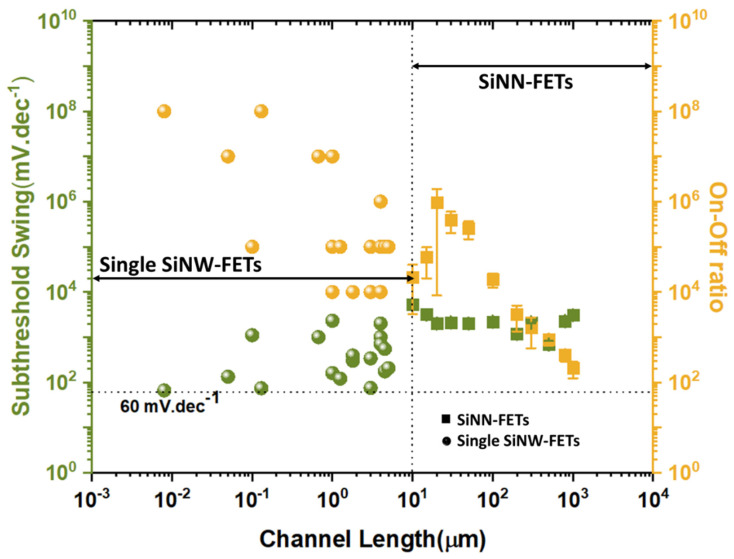
A comparison between single SiNW-FETs [11,33,175,181,182,183,184,185,186,187,188,189,190] and SiNN-FETs with respect to the channel length (SiNN density is about 0.6 NWs μm^−2^). Squares of SiNN-FETs segment show the average of several devices’ performances in which bars are representative of deviation in measured parameters for a certain *L_c_*.

**Figure 32 nanomaterials-12-01043-f032:**
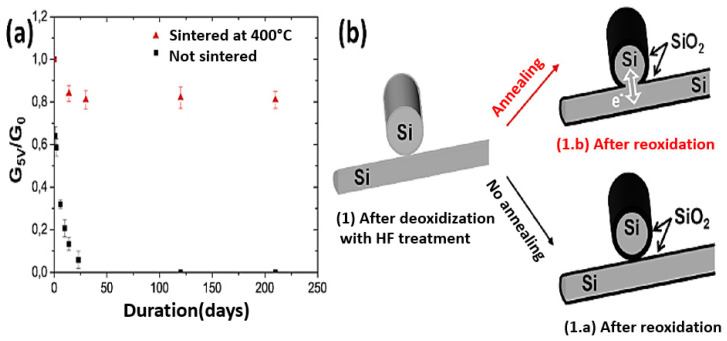
(**a**) Evolution of the conductance over time of Si nanonets based on not-annealed degenerated NWs and annealed at 400 °C after deoxidation. The conductance at 5 V was normalized to the initial conductance just after deoxidation. (**b**) Diagrams illustrating, after deoxidation, the reoxidation under air of an NW–NW junction (1.a) without annealing or (1.b) with annealing at 400 °C under nitrogen. Reproduced from [27].

**Figure 33 nanomaterials-12-01043-f033:**
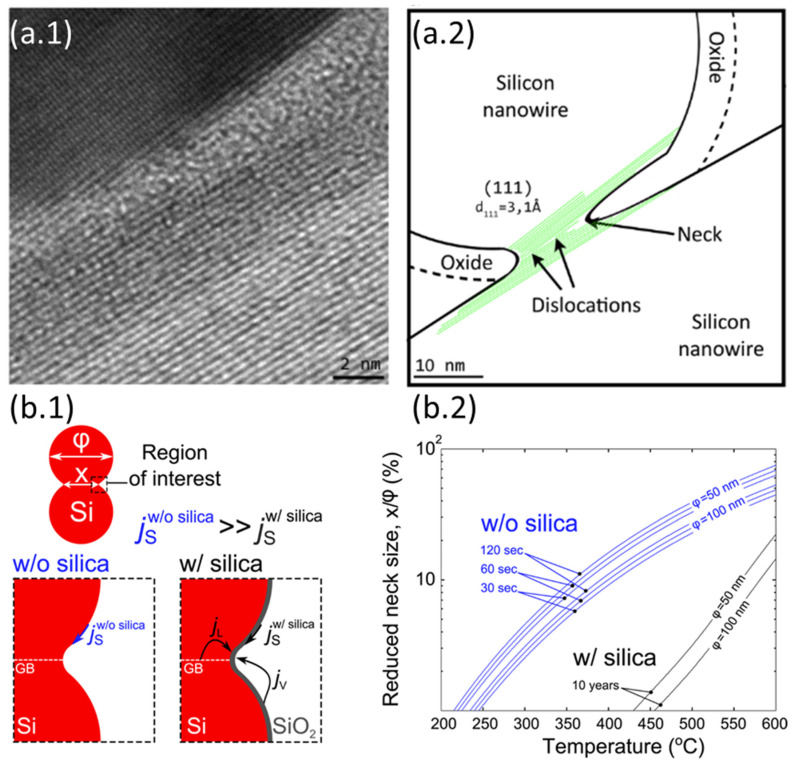
(**a.1**) High-resolution TEM image realized after re-oxidation of a junction between two NWs annealed at 400 °C under nitrogen. (**a.2**) Schematic representation of the TEM image showing the formation of a dislocation and a neck delimited by SiO_2_ at the NW–NW junction. (**b.1**) Modeling of the sintering between two silicon nanoparticles with (w/) and without (w/o) the native oxide. Surface diffusion (j_s_), volume diffusion from the grain boundary (j_L_), and vapor diffusion (j_v_) are the material transports considered. *X* and *φ* represent the shot size and diameter of the nanoparticles, respectively. (**b.2**) Sintering map representing the neck size relative to the initial nanoparticle size as a function of temperature for different annealing times. Reproduced from [48].

**Figure 34 nanomaterials-12-01043-f034:**
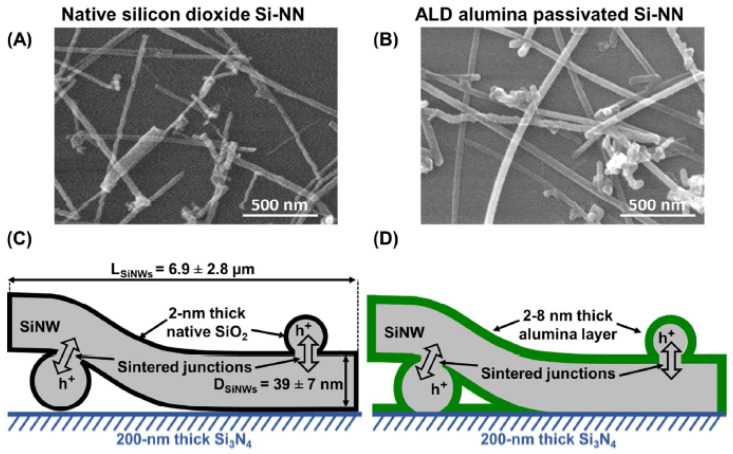
Comparison between sintered SiNN coated by (**A**,**C**) natively grown silicon dioxide and passivated by (**B**,**D**) alumina deposited using ALD. (**A**, **B**) Refer to top-view SEM images of nanonets, while (**C**, **D**) are sectional schemes of three coated SiNWs: one sectioned in the length and two according to the diameter. For (**C**), the mean and standard deviation of SiNW length (L_SiNWs_) and diameter (D_SiNWs_) are indicated. For (**D**), due to conformal coating with ALD, alumina is deposited simultaneously on SiNWs and onto the substrate whereas SiNW–SiNW junction and underneath SiNW portions are considered alumina-free. Reproduced from [47].

**Figure 35 nanomaterials-12-01043-f035:**
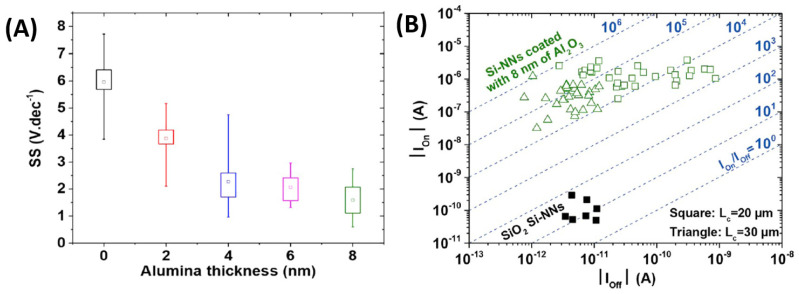
(**A**) Effect of the alumina thickness on the subthreshold slope (SS). 0 nm of alumina corresponds to a 2-nm thick layer of native SiO_2_. For all transistors, the channel length (*L_c_*) is 20 μm and the drain voltage (V_d_) was set at −4 V. The boxes show the 25th and 75th percentiles, whereas the whiskers represent the 5th and 95th percentiles. The empty square in the boxes shows the mean value. (**B**) Reproducibility of the on and off current for transistors based on native SiO_2_ SiNNs (full symbol) and 8-nm alumina encapsulated SiNNs (empty symbol) for 20 μm (square) and 30 μm (triangle) long channel. For native SiO_2_ SiNN based devices, no current is observed when the channel length is 30 μm. The on-to-off ratio (I_on_ = I_off_) is indicated by the dashed line. I_on_ and I_off_ were extracted at −25 V and +25 V, respectively. Reproduced from [47].

**Figure 36 nanomaterials-12-01043-f036:**
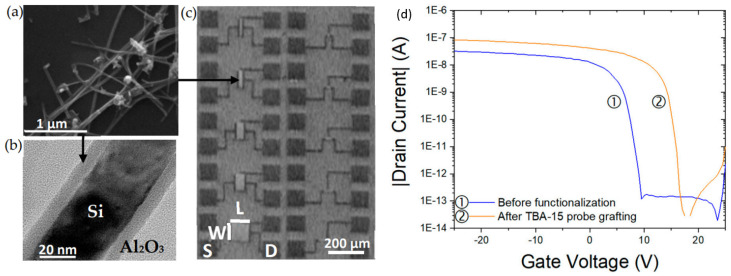
(**a**) SEM image of the Si nanonet (NN) after transfer on heavily doped Si substrate covered with a 200 nm thick Si_3_N_4_. (**b**) High-resolution TEM (HRTEM) image displaying an Al_2_O_3_ passivated Si nanowire. (**c**) Optical image of Al_2_O_3_ passivated SiNN field effect transistors (FETs) presenting different channel geometries. (**d**) Transfer characteristics were obtained for an L = 100 μm, W = 100 μm NN-FET at a drain voltage of V_ds_ = −2 V before and after functionalization with thrombin-binding aptamer (TBA-15). Reproduced from [200].

**Figure 37 nanomaterials-12-01043-f037:**
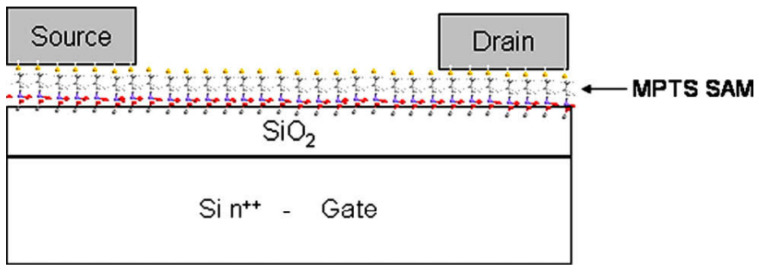
Schematic structure of the devices with MPTS as anchoring layer for gold electrodes. Reproduced from [201].

**Figure 38 nanomaterials-12-01043-f038:**
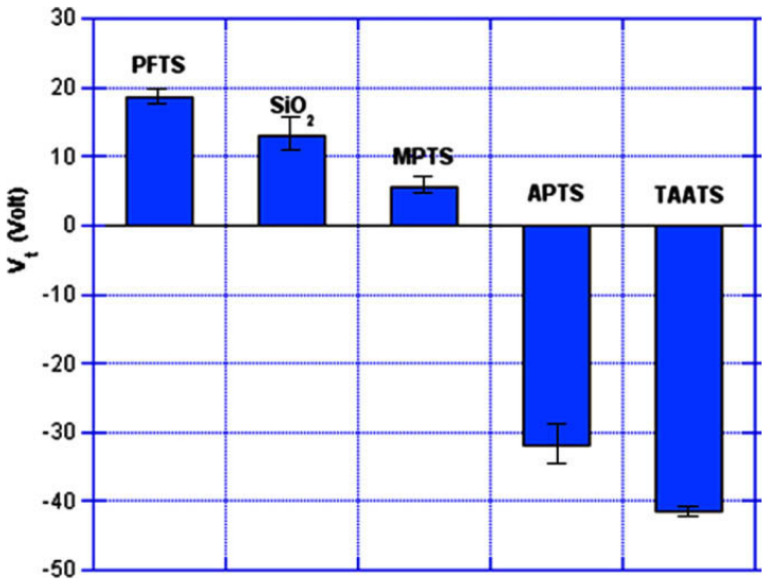
V_t_ as a function of the channel modification. Reproduced from [201].

**Figure 39 nanomaterials-12-01043-f039:**
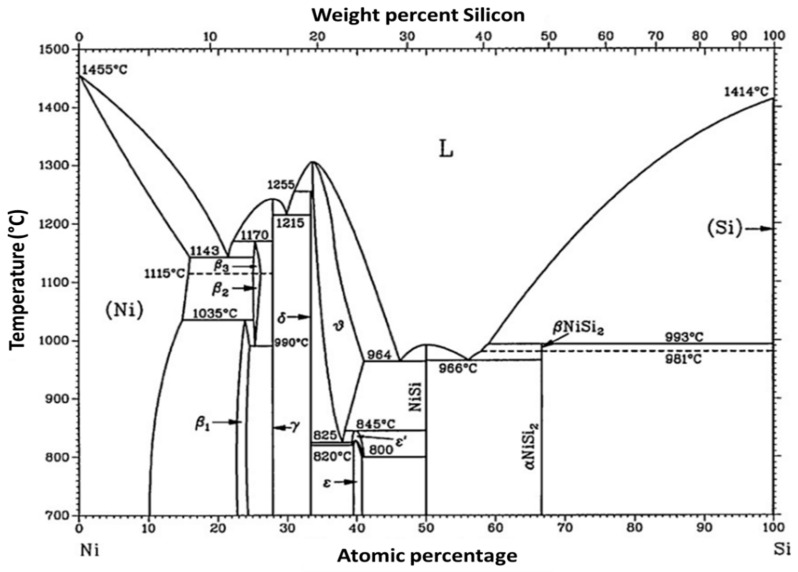
Binary phase diagram of the nickel–silicon pair. Reproduced from [209].

**Figure 40 nanomaterials-12-01043-f040:**
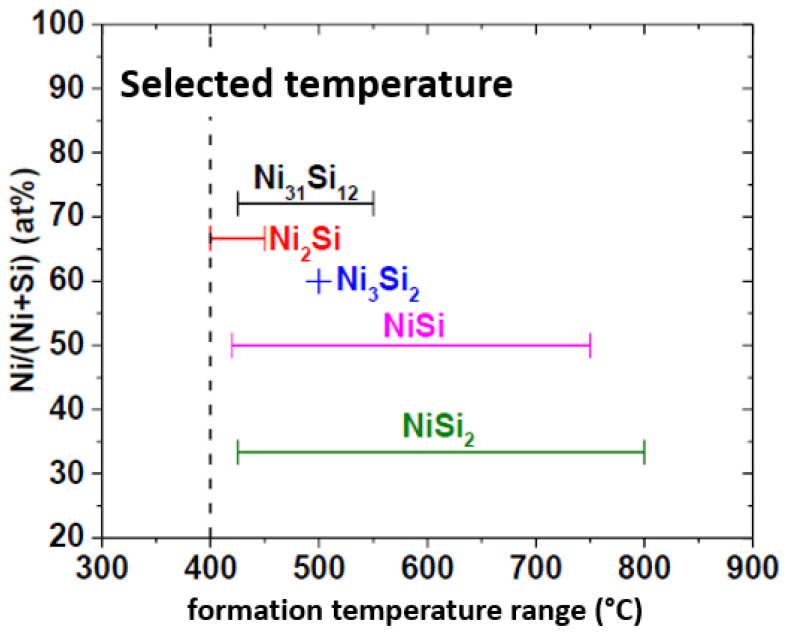
Temperature range of formation of different Ni_X_Si_y_ silicides in SiNWs reported in the literature from 15 references [175,176,182,213,214,216,217,218,219,220,221,222,223,224].

**Figure 41 nanomaterials-12-01043-f041:**
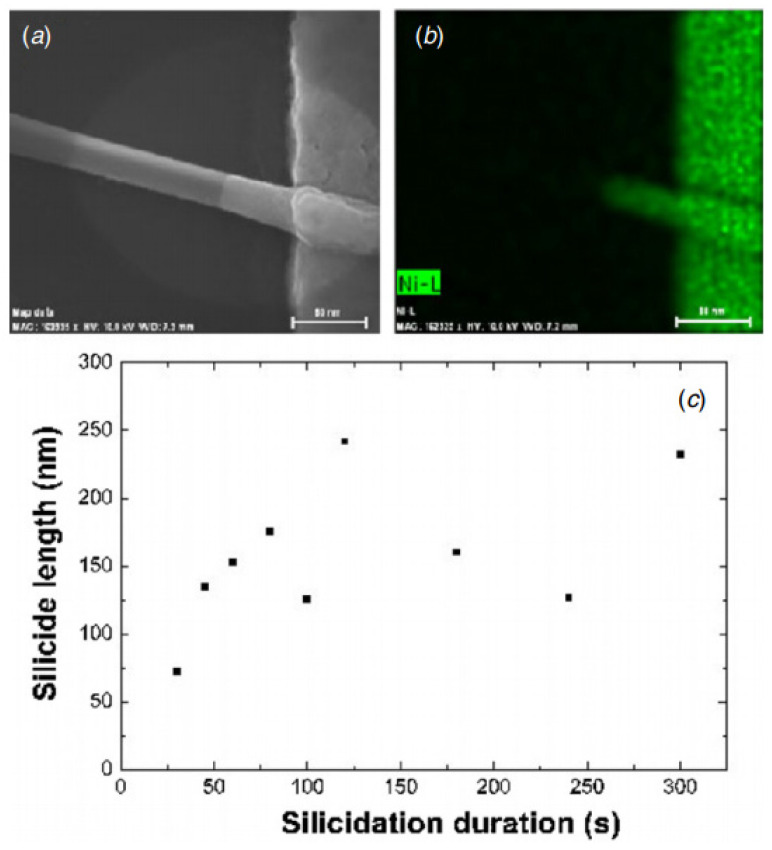
(**a**) SEM view of a silicided contact after annealing at 400 °C under nitrogen gas during 300 s. The scale bar is 80 nm. (**b**) EDX picture of the contact presented in (**a**). Green color indicates the presence of nickel. One can notice the propagation of the nickel in the SiNW after the annealing step leading to the formation of a silicide. (**c**) Length of the silicided section obtained at 400 °C under nitrogen gas in an RTA furnace as a function of the annealing time. Reproduced from [11].

**Figure 42 nanomaterials-12-01043-f042:**
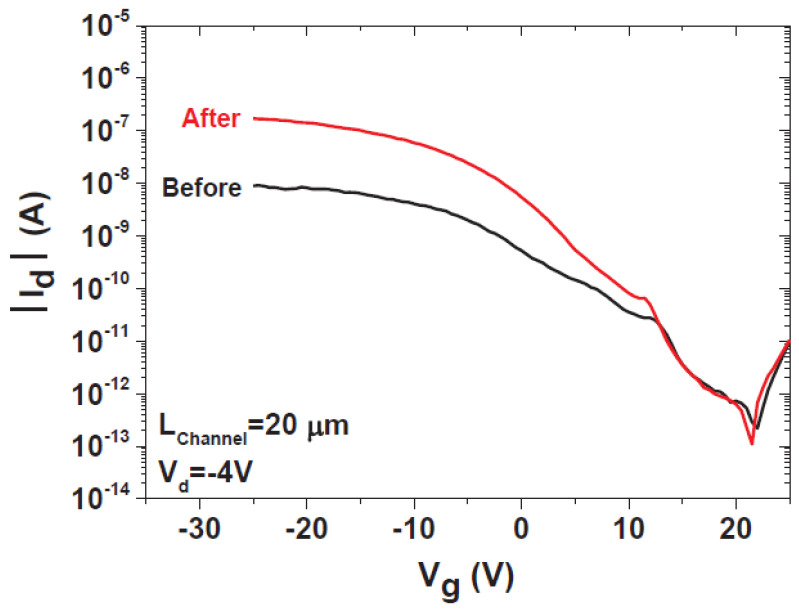
Transfer characteristics before and after silicidation (400 °C, for 60 s) of a 20 μm channel Si nanonet FET elaborated with 18 mL of filtered SiNW solution (with 42 × 10^6^ NWs cm^−2^ corresponding density). The drain-source bias was set at −4 V. Reproduced from [226].

**Figure 43 nanomaterials-12-01043-f043:**
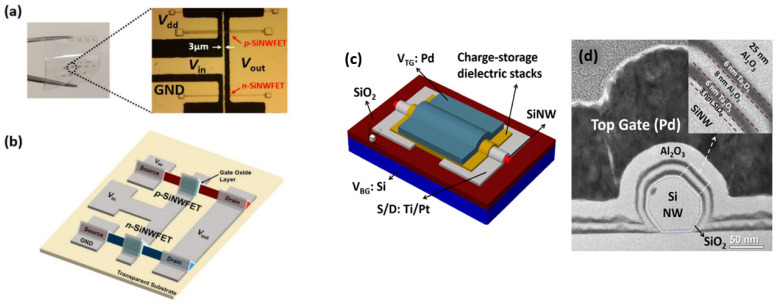
(**a**) Optical images and (**b**) schematic illustration of the SiNW CMOS inverter on a transparent substrate. Reproduced from [234]. (**c**) Schematic of a SiNW-FET-based charge-trapping non-volatile flash memory; (**d**) TEM image of the cross section of a MATATOS device. Inset demonstrates the typical thickness of the top gate stack. Reproduced from [231].

**Figure 44 nanomaterials-12-01043-f044:**
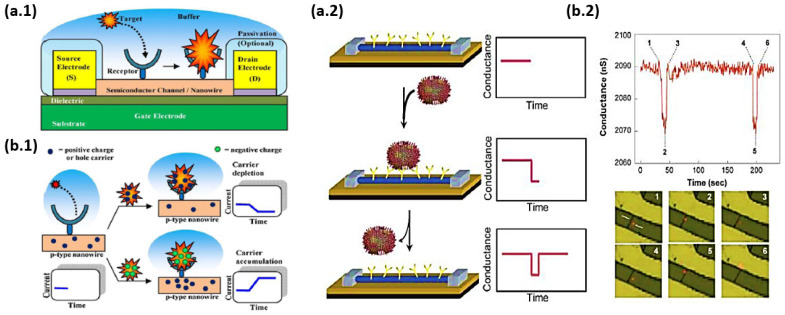
(**a.1**) The illustration of a nanoscale FET biosensor with a cross-sectional view. (**b.1**) When positively charged target molecules bind the receptor modified on a p-type NW, positive carriers (holes) are depleted in the NW, resulting in a decrease in conductance. Conversely, negatively charged target molecules captured by the receptor would make an accumulation of hole carriers, causing an increase in conductance. Reproduced from [174]. (**a.2**) Schematic of a single virus binding and unbinding to the surface of a SiNW device modified with antibody receptors and the corresponding time-dependent change in conductance. (**b.2**) Simultaneous conductance and optical data recorded for a Si nanowire device after the introduction of influenza A solution. The images correspond to the two binding/unbinding events highlighted by time points 1–3 and 4–6 in the conductance data, with the virus appearing as a red dot in the images. Reproduced from [17].

**Figure 45 nanomaterials-12-01043-f045:**
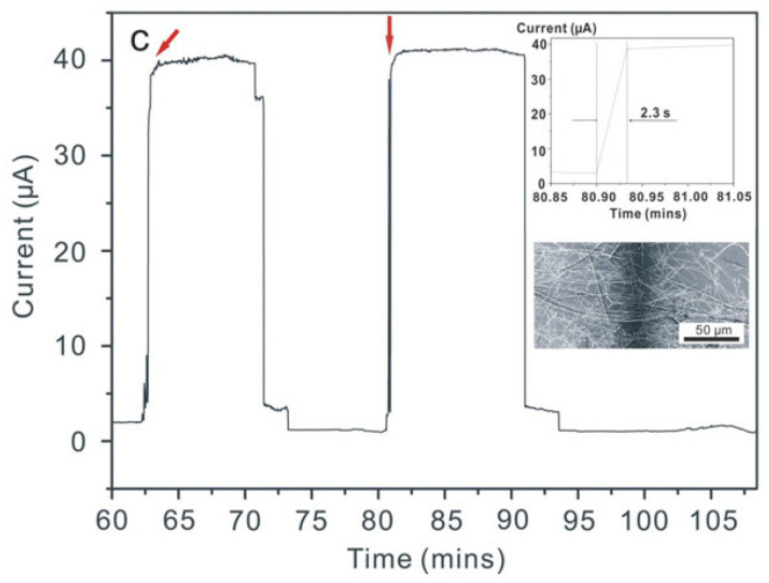
Real time current response of Pd coated SiNWs in 5% H_2_. The inset images show the enlarged current response (**upper**) and an SEM image (**below**) of the device. Note that, the sensor was inside a chamber with a pressure of 0.01 Torr, and a voltage of 2 V was applied across it. Reproduced from [236].

**Table 1 nanomaterials-12-01043-t001:** Comparison between different NW growth methods including low pressure VLS (LP VLS), high pressure VLS (HP VLS), SLS, and electrochemical.

Method	Diameter Range	Crystallinity	Doping	Yield	Scalability	Homogeneity	Catalyst	Freestanding
LP VLS	>30 nm	Monocrystalline	Highly controlled	Medium	Good	Generally dispersed	Yes	No
HP VLS	<30 nm	Polycrystalline	Controlled	High	Very good	Homogeneous	Yes	Yes
SLS	Between 20 and 30 nm	Monocrystalline	Controlled	Medium	Good	Dispersed	Yes	No
Electrochemical	Limited by template	Polycrystalline	Controlled	High	Good	Highly homogeneous	Yes	No

**Table 2 nanomaterials-12-01043-t002:** Advantages and disadvantages of each NWs transferring technique.

Technique	NWs Uniformity	Random/Aligned	Density (Percolation Regime)	NWs Layer Thickness (Low Density)	Versatile in Substrate	Scalability	Localized/Large Scale	Complexity
Drop-casting	Low	Random/Aligned	Low	Low	High	Low	Localized	Low
Fluidic directed	Medium	Aligned	Low	Low	Low	Low	Localized	Medium
Langmuir-Blodgett	High	Aligned	Medium	High	High	Medium	Localized/Large scale	Medium
Blown-bubble	High	Aligned	Medium	Low	High	Low	Localized	
Contact printing	Medium	Aligned	Medium	Medium	Medium	Low	Localized	Medium
Vacuum filtration	High	Random	High	High	Medium	Medium	Large scale	Low
Spray coating	High	Random/Aligned	Medium	Low	High	High	Large scale	Low

**Table 3 nanomaterials-12-01043-t003:** Properties of different silicides used in the microelectronics industry. Table adapted from references [205,207,208]. Φ_bh_ corresponds to the barrier height on N-type silicon.

Silicide	Formation Temperature (°C)	Crystalline Structure	Resistivity (μΩ.cm)	Φ_bh_ (eV)
TiSi_2_	650	Orthorhombic	13–16	0.60
CoSi_2_	450	Cubic	18–20	0.64
PtSi	300	Orthorhombic	28–35	0.87
NiSi	400	Orthorhombic	10.5–18	0.75

**Table 4 nanomaterials-12-01043-t004:** Formation temperature, crystallographic structure, electrical resistivity, unit cell volume per Si atom ($V_{Ni_{x}Si_{i}/Si}$), and the ratio of this volume to that of Si ($V_{Ni_{x}Si_{i}/Si}/V_{Si}$) of each room temperature stable Ni_X_Si_y_ silicide [205,208]. UNK is the acronym for unknown.

Phase	Formation Temperature (°C)	Crystalline Structure	Resistivity (μΩ.cm)	$V_{Ni_{x}Si_{i}/Si}\;({Å}^{3})$	$V_{Ni_{x}Si_{i}/Si}/V_{Si}$
Ni	-	Cubic	7–10	-	-
Ni_3_Si	UNK	Cubic	80–90	43.08	2.15
Ni_31_Si_12_	UNK	Hexagonal	90–150	39.46	1.97
Ni_2_Si	200	Orthorhombic	24–30	32.15	1.61
Ni_3_Si_2_	UNK	Orthorhombic	60–70	28.73	1.44
NiSi	400	Orthorhombic	10.5–18	24.12	1.21
NiSi_2_	800	Cubic	34–50	19.75	0.99
Si	-	Cubic	Depend on doping	20.01	1

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
