# Peer review of "Functional Devices from Bottom-Up Silicon Nanowires: A Review"

_nanomaterials, 2022, doi:10.3390/nano12071043_

Round 1

Reviewer 1 Report

This review summarises the work related to the bottom-up Si NWs. It is an important to do this work, as the Si NW is highly useful for both research and practical applications. This work can provide useful and uptodate information to researchers. I am impressed by the careful and complete summary of the field by authors.  Thus, I would like to accept it for publication. 

Author Response

We thank the reviewer for his/her positive evaluation of our work.

A careful review of the paper has been carried out to check English and we hope to have now solved the question.

Reviewer 2 Report

These review paper aims to provide an overview of the recent and ongoing research efforts in the field of bottom-up silicon nanowires. In general, the manuscript addresses an interesting and timely topic and it is worth to be published. The paper is well structured and well written although I suggest several minor corrections to ease the reading and improve its completeness.

1) Figure 3 show two different kind of nanonet structures. 3D and 2D though the paper just includes applications based on single nanowires and 2D systems.  A description of possible applications of 3D structures would certainly help to improve the review completeness.

2) The description of the field-effect transistor-based biosensor performed in section 3.2 would be better fit on section 7 where the details of different FET devices and their performance are described.

3) Figure 7 shows that the dispersion current of single NWs is much higher than that of Nanonet structures. This performance improvement however occurs with the counterpart of having much less current.  A comment on this point and its main drawbacks towards applications should be provided.

4) The description provided in section 4.5 “Possibility of functionalization” is very poor.  This part should be included when described biological sensors or a more developed description may be performed if it is part of a full section.

5) A summary with a comparison of properties, sizes, advantages/disadvantages of nanowires growth with different techniques may be provided in section 5.

6) the process schematically shown in Figure 10 (A) using functionalized patterned substrates is not described in the text.

7) Part of the description performed in caption of Figure 15 would better fit in the text.

8) Drop-casting process is not described within the different nanowire transferring techniques.

9) Chapters described in the abstract are not correct.

Reviewer 3 Report

This paper summarizes the main aspects of the Si nanowires research progress, from the bottom-up growth to their applications in electronic devices. The review is exhaustive and well organized. I can recommend publication in the special issue "silica and silicon based nanostructures" of Nanomaterials.

I have only few comments/suggestions:

1) In the abstract the mumber of sections looks incorrect (it does not correspond to the section numbering in the index). Moreover the page numbering in the table of content is wrong.

2) Among the many peculiar properties of semiconductor nanowires, one of the most attractive is the strain relaxation that gives the possibility to combine also lattice mismatch materials in axial heterostructure without the formation of misfit dislocations. Even if this review focusses on pure Si nanowires, nanowire heterostructures (like Si/Ge) have shown very interesting properties opening new possibilities in material engineering. This important aspect is missing ant it should be mentioned in section 3.
